# Intellectual disability-associated disruption of O-GlcNAc cycling impairs habituation learning in *Drosophila*

**Michaela Fenckova**[1,2,☯,¤a]*, **Villo Muha**[1,☯], **Daniel Mariappa**[1,☯,¤b], **Marica Catinozzi**[3], **Ignacy Czajewski**[1], **Laura E. R. Blok**[2], **Andrew T. Ferenbach**[1], **Erik Storkebaum**[3], **Annette Schenck**[2,‡], **Daan M. F. van Aalten**[1,‡]*

**1** Division of Gene Regulation and Expression, School of Life Sciences, University of Dundee, Dundee, United Kingdom, **2** Department of Human Genetics, Donders Institute for Brain, Cognition and Behaviour, Radboud University Medical Center, Nijmegen, Netherlands, **3** Department of Molecular Neurobiology, Donders Institute for Brain, Cognition and Behaviour, Faculty of Science, Radboud University, Nijmegen, Netherlands

☯ These authors contributed equally to this work.
¤a Current address: Department of Molecular Biology and Genetics, Faculty of Science, University of South Bohemia in Ceske Budejovice, Ceske Budejovice, Czechia
¤b Current address: Drosophila Genomics Resource Center, Biology Department, Indiana University, Bloomington, Indiana, United States of America
‡ shared last authors.
* fenckm00@prf.jcu.cz (MF); d.m.f.vanaalten@dundee.ac.uk (DMFA)

**Data Availability Statement:** All relevant data are within the manuscript and its Supporting Information files.

## Abstract

O-GlcNAcylation is a reversible co-/post-translational modification involved in a multitude of cellular processes. The addition and removal of the O-GlcNAc modification is controlled by two conserved enzymes, O-GlcNAc transferase (OGT) and O-GlcNAc hydrolase (OGA). Mutations in *OGT* have recently been discovered to cause a novel Congenital Disorder of Glycosylation (OGT-CDG) that is characterized by intellectual disability. The mechanisms by which OGT-CDG mutations affect cognition remain unclear. We manipulated O-GlcNAc transferase and O-GlcNAc hydrolase activity in *Drosophila* and demonstrate an important role of O-GlcNAcylation in habituation learning and synaptic development at the larval neuromuscular junction. Introduction of patient-specific missense mutations into *Drosophila* O-GlcNAc transferase using CRISPR/Cas9 gene editing leads to deficits in locomotor function and habituation learning. The habituation deficit can be corrected by blocking O-GlcNAc hydrolysis, indicating that OGT-CDG mutations affect cognition-relevant habituation via reduced protein O-GlcNAcylation. This study establishes a critical role for O-GlcNAc cycling and disrupted O-GlcNAc transferase activity in cognitive dysfunction, and suggests that blocking O-GlcNAc hydrolysis is a potential strategy to treat OGT-CDG.

## Author summary

Attachment of single N-acetylglucosamine (GlcNAc) sugars to intracellular proteins has recently been linked to neurodevelopment and cognition. This link has been strengthened

**Funding:** This work was supported by a ZonMW Vici grant from the Nederlandse Organisatie voor Wetenschappelijk Onderzoek (nwo.nl, grant no. 09150181910022, M.F. received salary from this grant) and the Australian National Health & Medical Research Council Centre for Research Excellence Scheme (www.nhmrc.gov.au, grant no. APP1117394, L.E.R.B received salary from this grant) to A.S., by an h2020 European Research Council (erc.europa.eu) consolidator grant (grant no. ERC-2017-COG 770244, M.C. and E.S. received salary from the grant) to E.S., by a mobility grant from Ministersvo Školství, Mládeže a Tělovýchovy (www.msmt.cz, grant no. CZ.02.2.69/0.0/0.0/ 20_079 /0017633, M.F. received salary from this grant) to M.F. and by a Wellcome Trust Investigator Award (wellcome.org, grant no. 110061) and the National Centre for the Replacement, Refinement and Reduction of Animals in Research (https://www.nc3rs.org.uk/; grant no. T001682) to D.M.F.A. The funders had no role in study design, data collection and analysis, decision to publish, or preparation of the manuscript.

**Competing interests:** The authors have declared that no competing interests exist.

by discovery of O-GlcNAc transferase (OGT) missense mutations in intellectual disability. Most of these mutations lie outside the catalytic O-GlcNAc transferase domain and it is unclear how they affect cognitive function. Using the fruit fly *Drosophila melanogaster* as a model organism, we found that a balance in O-GlcNAc cycling is required for learning and neuronal development. Habituation, a fundamental form of learning, is affected in flies that carry patient-specific OGT mutations and increasing O-GlcNAcylation genetically corrects the habituation deficit. Our work establishes a critical role for O-GlcNAc cycling in a cognition-relevant process, identifies defective O-GlcNAc transferase activity as a cause of intellectual disability, and proposes underlying mechanisms that can be further explored as treatment targets.

## Introduction

O-GlcNAcylation is an essential and dynamic co-/posttranslational modification that is characterized by the attachment of an N-acetylglucosamine (GlcNAc) molecule to serine or threonine residues of intracellular proteins. O-GlcNAcylation is implicated in a wide range of cellular processes, such as: chromatin remodeling [1–3], transcription [4,5] and translation [6], Ras-MAPK and insulin signaling [7–9], glucose homeostasis [10], mitochondrial trafficking [11], and control of the circadian clock [12]. The addition and removal of the O-GlcNAc modification, termed O-GlcNAc cycling, is controlled by two evolutionarily conserved enzymes, O-GlcNAc transferase (OGT) and O-GlcNAc hydrolase (OGA).

OGT, responsible for the addition of O-GlcNAc, is abundantly expressed in neurons and is enriched in the postsynaptic density (PSD) [13], a protein-dense structure that organizes the postsynaptic signaling machinery. O-GlcNAcylation is altered in brains of patients with Alzheimer's disease [14] and animal and cellular models of major neurodegenerative diseases [15–19]. In *C. elegans* models of neurodegeneration, O-GlcNAcylation protects against neurotoxicity [20]. Furthermore, it plays an important role in neuronal regeneration through the synchronization of insulin signaling-dependent regenerative processes [8]. Recent studies also point to important functions in neuronal development, such as neuronal differentiation [21,22], assembly and axonal transport of neurofilaments [23], and synapse maturation [24]. Missense mutations in human *OGT* gene, located on the X chromosome, are associated with intellectual disability (ID) [25–31], a severe neurodevelopmental disorder that is characterized by impaired cognition. Patients with *OGT* mutations suffer from a wide array of clinical features, including intrauterine growth retardation, developmental delay, delayed or restricted language skills and severe learning difficulties. The syndrome has been termed OGT-associated Congenital Disorder of Glycosylation (OGT-CDG) [32]. These findings and animal studies suggest that O-GlcNAcylation plays an important function in cognitive processes, such as learning [33,34].

The OGT protein consists of an N-terminal tetratricopeptide (TPR) domain, which contributes to substrate recognition and binding [35] and a C-terminal catalytic domain that is responsible for glycosylation of the TPR-bound proteins. OGT-CDG mutations have been found in both domains. Unlike the mutations in the catalytic domain, the mutations in the TPR domain have not been shown to significantly affect global protein O-GlcNAc levels but they do affect the OGT-TPR domain substrate binding and glycosylation kinetics, as derived from *in vitro* assays and crystal structure analysis [26,36,37]. However, it remains to be known whether impaired glycosylation is the mechanism that leads to developmental and cognitive

defects caused by identified TPR domain mutations. This question is pertinent as the TPR domain has been shown to be essential for cellular functions other than glycosylation [38].

*Drosophila* as a model organism has contributed to understanding the disease pathogenesis of numerous (ID) syndromes. It offers a combination of well-established gene targeting approaches and a plethora of morphological and functional disease-relevant phenotypes. The *Drosophila OGT* orthologue is encoded by the polycomb group gene *super sex combs* (*sxc*) [39]. It is highly similar to human *OGT* [40]. Complete loss of *sxc* results in severe homeotic transformations of adult body structures and pupal lethality [41]. This lethality can be restored by ubiquitous over-expression of human *OGT*, demonstrating functional conservation [39]. sxc has also been associated with neuronal function in circadian rhythm regulation [42]. Therefore, *Drosophila* is highly suited to investigate the disrupted mechanisms underlying OGT-CDG.

Here we genetically manipulated both O-GlcNAc transferase and O-GlcNAc hydrolase activity using established *sxc* and *Oga* mutants and investigated the effect of O-GlcNAc cycling on cognitive function and neuronal development. We turned to habituation, an evolutionary conserved form of non-associative learning that is characterized by response decrement towards a repeated, non-meaningful stimulus. At the neuronal level, habituation is mediated by adaptive changes in the excitatory activity of the stimulus response pathway, causing attenuated downstream neuronal responses to repeated, familiar stimuli. Habituation thus serves as a filter mechanism that prevents information overload and allows cognitive resources to focus on relevant stimuli [43]. It represents a prerequisite for higher cognitive functions [44–46]. Deficits in habituation have been reported in a number of neurodevelopmental disorders [47] and in more than a hundred *Drosophila* ID models [48,49]. Synaptic morphology at the *Drosophila* neuromuscular junction, an established model synapse [50], was assessed as a measure of neuronal development. To independently validate our findings we targeted the endogenous *sxc* locus by CRISPR/Cas9 editing [51] and generated a strong hypomorph mutation in the O-GlcNAc transferase domain (*sxc*$^{H596F}$). With the same technique we introduced three ID-associated missense mutations found in the conserved TPR domain of OGT: R284P [28], A319T [25] and L254F [29] and generated equivalent *sxc*$^{R313P}$, *sxc*$^{A348T}$ and *sxc*$^{L283F}$ alleles, respectively. We evaluated their effect on protein O-GlcNAcylation, developmental viability, adult lifespan and locomotor activity, and assessed their effect on cognitive function and neuronal development.

We find that appropriate O-GlcNAc cycling is required for habituation, a fundamental form of learning that is widely disrupted in *Drosophila* models of ID, and for synaptic development. We show that *OGT* missense mutations, implicated in ID, outside the catalytic O-GlcNAc transferase domain lead to deficits in habituation learning and that these deficits are caused by disrupted O-GlcNAc transferase activity. We thus unambiguously demonstrate the role of O-GlcNAcylation in brain development and function.

## Results

### Alteration of O-GlcNAc transferase activity leads to a deficit in habituation

To investigate whether the catalytic activity of *Drosophila* OGT is required for cognitive function, we tested the effect of an *sxc* mutation with diminished catalytic activity, *sxc*$^{H537A}$, on habituation. The homozygous hypomorphic flies are viable and except mild wing vein and scutellar bristle phenotypes do not present with any morphological abnormalities [51]. In habituation, an initial strong response towards a repeated but harmless stimulus gradually wanes based on prior experience. To assess this phenotype we used light-off jump habituation, an established non-associative learning assay that meets the strict habituation criteria, including spontaneous recovery and dishabituation with novel stimulus and excluding sensory adaptation and motor fatigue [46,52]. We subjected *sxc*$^{H537A}$ homozygous (*sxc*$^{H537A/H537A}$),

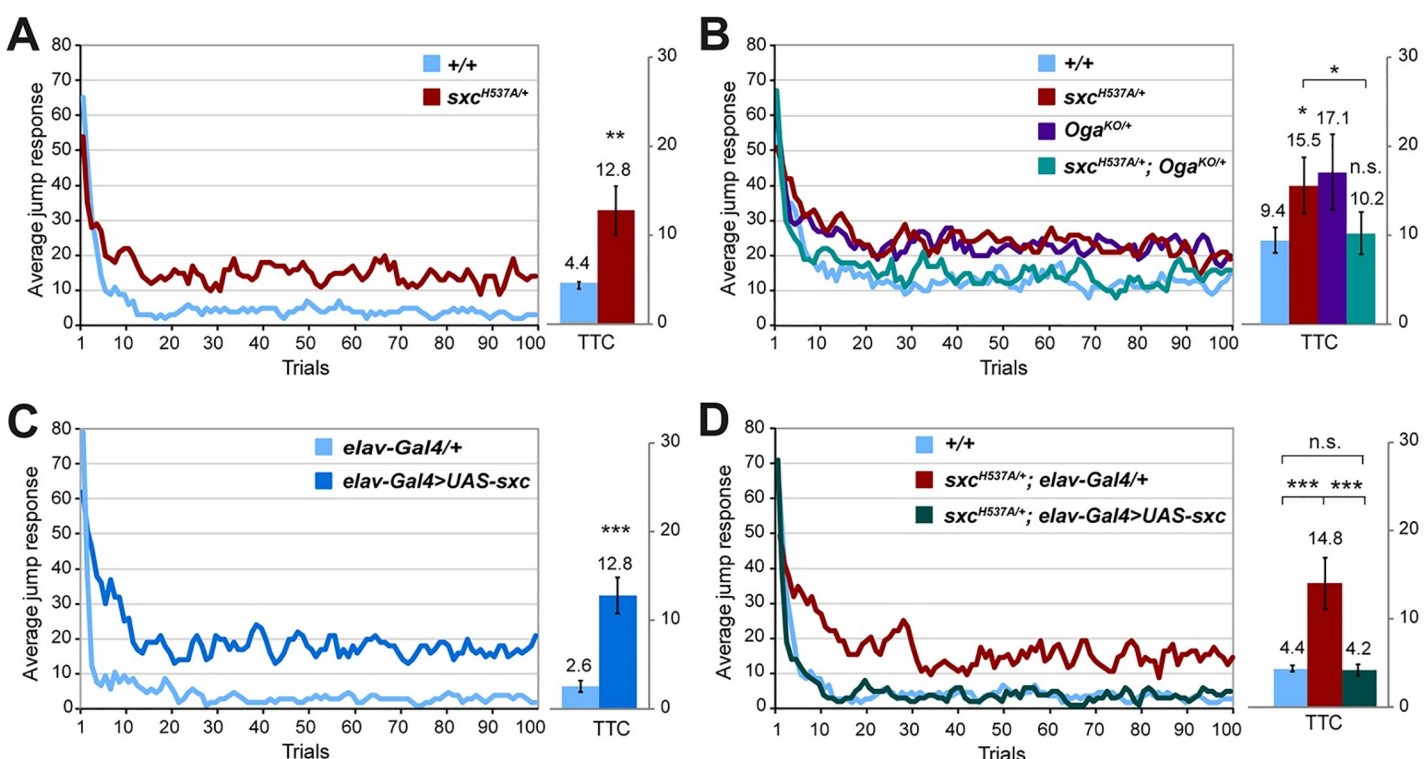

**Fig 1. Catalytic activity of O-GlcNAc transferase in neurons is required for habituation learning.** Jump responses were induced by 100 light-off pulses with 1 s intervals between pulses. The jump response represents the % of jumping flies in each light-off trial. The mean number of trials that flies needed to reach the no-jump criterion (Trials To Criterion, TTC) ± SEM is also shown. (A) Defective habituation of $sxc^{H537A/+}$ flies (N = 59, mean TTC ± SD: 12.8 ± 6.7 $p$ = 0.001, in red) compared to their respective genetic background control flies (+/+, mean TTC ± SD: 4.4 ± 0.9, N = 61, in blue). $^{**}$ $p$<0.01, based on lm analysis. (B) Habituation defect of $sxc^{H537A/+}$ flies (N = 72, mean TTC ± SD: 15.5 ± 7.8 $p_{adj}$ = 0.045, in red) is corrected by removing one *Oga* allele in $sxc^{H537A/+}$; $Oga^{KO/+}$ flies (N = 70, mean TTC ± SD: 10.1 ± 5.9, $p_{adj}$ = 0.024, in cyan) to the level of control flies (N = 72, mean TTC ± SD: 9.4 ± 3.5, $p_{adj}$ = 0.677, in blue). Habituation of $Oga^{KO/+}$ flies (N = 76, mean TTC ± SD: 17.1 ± 10.4, in purple) is slower but not significantly different from the control flies ($p_{adj}$ = 0.467). (C) Defective habituation of *elav-Gal4>UAS-sxc* flies (N = 55, mean TTC ± SD: 12.8 ± 4, $p_{adj}$ = 3.89x10$^{-12}$, in dark blue) compared to control *elav-Gal4/+* flies (N = 38, mean TTC ± SD: 2.6 ± 1.3 in light blue). (D) Habituation defect of $sxc^{H537A/+}$; *elav-Gal4/+* flies (N = 40, mean TTC ± SD: 14.6 ± 6, $p_{adj}$ = 4.92x10$^{-12}$, in red) is corrected by selective expression of *UAS-sxc* in neurons ($sxc^{H537A/+}$; *elav-Gal4>UAS-sxc*, N = 43, mean TTC ± SD: 4.2 ± 1.3, $p_{adj}$ = 8.94x10$^{-7}$, in green), to the level of the genetic background control flies (+/+, N = 65, mean TTC ± SD: 4.4 ± 0.9, $p_{adj}$ = 0.68, in blue). $^{*}$ $p_{adj}$<0.1, $^{***}$ $p_{adj}$<0.001, n.s. not significant, based on lm analysis with Bonferroni-Holm correction for multiple comparisons. A complete list of p-values and summary statistics is provided in S3 Table.

heterozygous ($sxc^{H537A/+}$), and genetic background control flies (+/+) to 100 light-off pulses in the light-off jump habituation assay. While $sxc^{H537A/+}$ and control flies exhibited good initial jump responses to the light-off stimuli (61% and 67% initial jumpers out of N = 96 tested flies per genotype; above a required threshold of 50% [49]), $sxc^{H537A/H537A}$ flies were impaired (36% initial jumpers, N = 64), identifying broader defects that preclude assessment of habituation. Compared to control flies that habituated quickly to the repeated light-off stimulus, $sxc^{H537A/+}$ flies displayed significantly slower habituation and needed significantly more light-off pulse trials to suppress their jump response (Trials To no-jump Criterion, TTC). Some mutant flies were not able to suppress their jump response during the entire course of the experiment, as reflected by the high baseline of the average jump response curve (**Fig 1A**). These results suggest that partial loss of O-GlcNAc transferase activity or altered O-GlcNAcylation kinetics in $sxc^{H537A/+}$ mutants impairs habituation.

We validated our conclusion by employing a knockout-out allele of *Drosophila Oga* [53]. We asked whether partial inhibition of O-GlcNAc hydrolysis, by removing one copy of *Oga* ($Oga^{KO/+}$) could improve habituation of the $sxc^{H537A/+}$ flies. Habituation of $Oga^{KO/+}$ flies was also slower but not significantly different from the control flies. The transheterozygous

*sxc*<sup>H537A/+</sup>; *Oga*<sup>KO/+</sup> flies showed good initial jump responses, and their habituation was not significantly different to that of control flies, identifying a significant improvement compared to *sxc*<sup>H537A/+</sup> flies (**Fig 1B**). The fatigue assay (see Materials and Methods) confirmed that the lower TTC values were not a result of increased fatigue (**S2A Fig**). These results show that *Drosophila* is a suitable model to study the role of OGT in cognitive functioning and demonstrate that tight control of protein O-GlcNAcylation is required for proper habituation learning in *Drosophila*.

## O-GlcNAc transferase is required for habituation in neurons

We next sought to determine whether the *sxc*<sup>H537A/+</sup> habituation deficit originates from reduced OGT function in neurons. We therefore induced neuronal knockdown of *sxc* by crossing the pan-neuronal elav-Gal4 driver line (see Materials and Methods) to an inducible RNAi line against *sxc* obtained from Vienna *Drosophila* Resource Center *(#18610*, zero predicted off-targets). Progeny from crossing the driver line to the isogenic genetic background of the RNAi line *(#60000)* were used as controls. While control flies (*elav-Gal4/+*) showed good initial jump response (56%, N = 64), *elav-Gal4>UAS-sxc*<sup>RNAi</sup> flies—similar to *sxc*<sup>H537A/H537A</sup> flies—exhibited very low initial jump response to light-off stimulus (19% initial jumpers, N = 96). While this detrimental effect prevented assessment of *sxc* neuron-specific knockdown in habituation learning, it does argue that i) the failed jump response of *sxc*<sup>H537A/H537A</sup> flies is likely due to loss of OGT activity in neurons, and ii) OGT activity is indispensable for basic neuronal function or neuronal development.

We used an alternative strategy to test whether habituation deficits of *sxc*<sup>H537A/+</sup> flies are of neuronal origin. We asked whether restoration of OGT activity in neurons can correct habituation deficits of *sxc*<sup>H537A/+</sup> flies, by inducing pan-neuronal overexpression of functional wild-type sxc in the heterozygous *sxc*<sup>H537A/+</sup> as well as control background. Neuronal overexpression of functional sxc in control flies (*elav-Gal4>UAS-sxc*) resulted in a habituation deficit (**Fig 1C**). This is consistent with our previous findings of habituation deficits in *Oga*<sup>KO/KO</sup> flies, which also show increased protein O-GlcNAcylation [53]. In contrast, re-expression of functional *sxc* in the *sxc*<sup>H537A/+</sup> flies completely corrected their habituation deficits (**Fig 1D**). The lower TTCs were not a result of increased fatigue (**S2B Fig**). Therefore, appropriate levels of OGT activity and O-GlcNAcylation, specifically in neurons, are required for habituation learning.

## Neuronal O-GlcNAc transferase activity controls synaptic development

Synapse biology is important for brain development and cognition, and abnormalities in synaptic architecture are characteristic of multiple *Drosophila* models of neurodevelopmental disorders [54–59]. For these reasons, we asked whether *sxc*<sup>H537A/+</sup> mutants show any defects in the morphology of the third instar larval neuromuscular junction (NMJ), a well-established model synapse. We labeled NMJs by immunostaining for the postsynaptic membrane marker anti-discs large (Dlg1) to visualize the overall morphology of the NMJ terminal, and for synaptotagmin (Syt), a synaptic vesicle marker that visualizes synaptic boutons. We did not observe a significant change in synaptic length, area, or perimeter (**Fig 2A**), nor in the number of branches and branching points (**S3A Fig**) in NMJs of *sxc*<sup>H537A/+</sup> mutant larvae but we observed a significant increase in bouton number compared to the genetic background control (**Fig 2A**). Increasing OGT activity by presynaptic overexpression of wild-type *sxc* (*elav-Gal4>UAS-sxc*) resulted in an opposite phenotype: a decreased number of boutons compared to both controls, UAS transgene and driver alone. NMJ length was also reduced (**Fig 2B**). Both parameters were normalized when we neuron-specifically expressed functional *sxc* in the *sxc*<sup>H537A/+</sup>

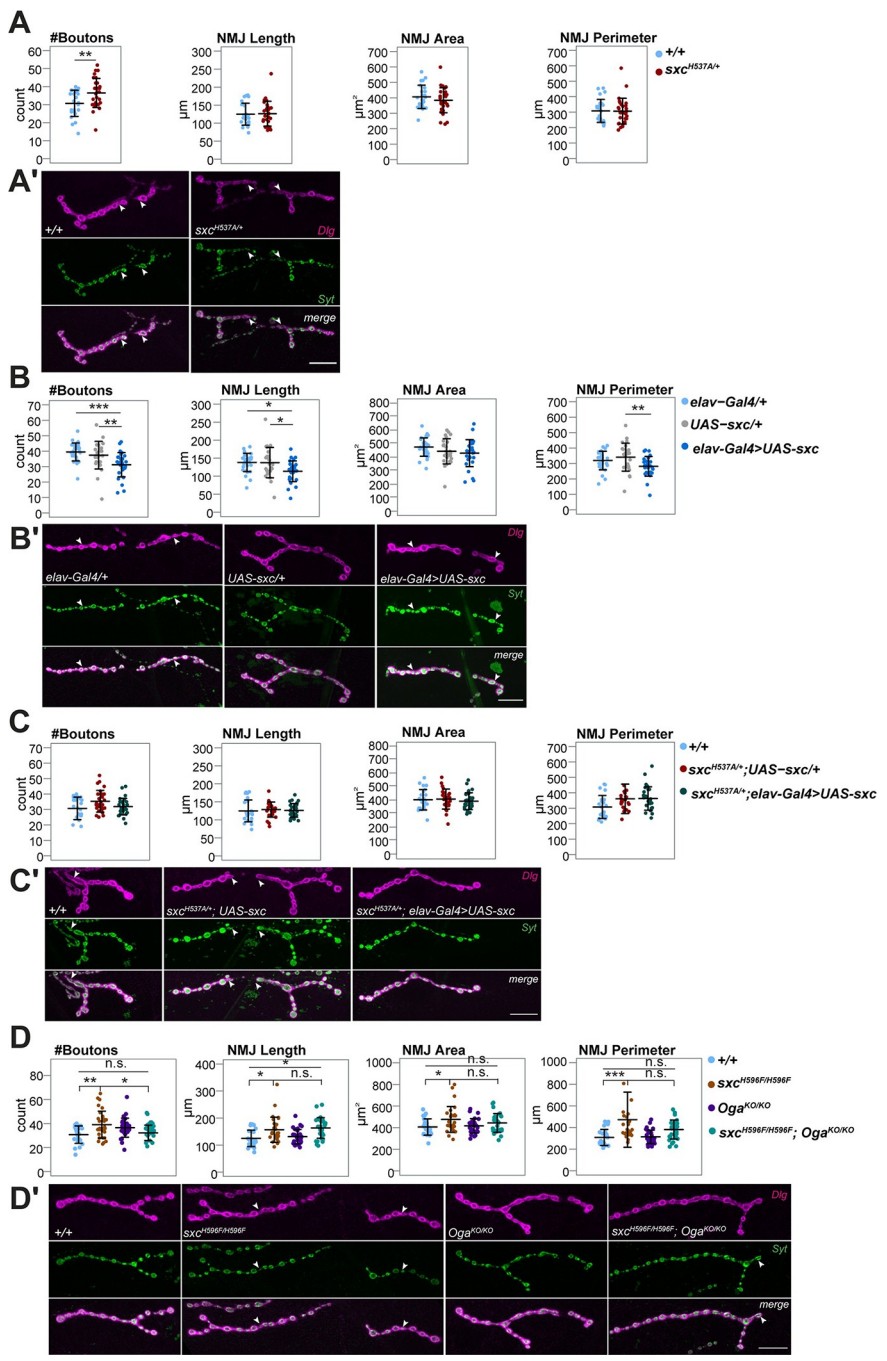

**Fig 2. Both reduced and increased O-GlcNAc transferase activity in neurons cause defects in synaptic morphology.** Data presented as individual data points with mean ± SD. (A) NMJs on muscle 4 of $sxc^{H537A/+}$ larvae have a significantly higher number of synaptic boutons (N = 29, in red) as compared to their genetic background control (+/+, N = 24, $p$ = 0.009, in blue,) but not significantly different NMJ length ($p$ = 0.872), area ($p$ = 0.314) and perimeter ($p$ = 0.935). $^{**}$ $p<0.01$, based on one-way ANOVA. (B) *Elav-Gal4>UAS-sxc* larvae have a significantly lower number of boutons (N = 29, in dark blue) compared to their respective background controls *elav-Gal4/+* (N = 30, $p_{adj}$ = 2.1x10$^{-4}$, in light blue) and *UAS-sxc/+* (N = 26, $p_{adj}$ = 0.009, in grey), significantly reduced NMJ length ($p_{adj/elav-Gal4}$ = 0.013, $p_{adj/UAS-sxc}$ = 0.02) and a smaller NMJ perimeter ($p_{adj/UAS-sxc}$ = 0.008). (C) Neuron-selective expression of sxc in $sxc^{H537A/+}$ larvae shows no change in bouton numbers ($sxc^{H537A/+}$; *elav-Gal4>UAS-sxc* (N = 29, in green)), compared to $sxc^{H537A/+}$, *UAS-sxc/+* larvae (N = 28, $p_{adj}$ = 0.102, in red) and compared to control (N = 27, $p_{adj}$ = 0.977, in blue) and no change in NMJ length compared to control (+/+, N = 28, $p_{adj}$ = 0.974) and $sxc^{H537A/+}$, *UAS-sxc/+* larvae ($p_{adj}$ = 0.935). Also NMJ area ($p_{adj/control}$ = 0.849, $p_{adj/sxcH537A; UAS-sxc/+}$ = 0.691) and NMJ perimeter were not changed ($p_{adj/control}$ = 0.066), $p_{adj/sxcH537A; UAS-sxc/+}$ = 0.995). (D) Number of synaptic boutons ($p_{adj}$ = 0.003), NMJ length ($p_{adj}$ = 0.01),

area ($p_{adj}$ = 0.028) and perimeter ($p_{adj}$ = 4.5x10$^{-4}$) are significantly increased in $sxc^{H596F/H596F}$ larvae (N = 31, in brown) compared to the genetic background control larvae (+/+, N = 28, in blue) and partially normalized in the $sxc^{H596F/H596F}$; $Oga^{KO/KO}$ larvae (N = 30, in cyan, boutons: $p_{adj/sxcH596F}$ = 0.011, $p_{adj/control}$ = 0.923; length: $p_{adj/sxcH596A}$ = 0.896, $p_{adj/control}$ = 0.001; area: $p_{adj/sxcH596A}$ = 0.501, $p_{adj/control}$ = 0.431; perimeter: $p_{adj/sxcH596A}$ = 0.08, $p_{adj/control}$ = 0.26). None of the parameters is significantly affected in the $Oga^{KO}$ larvae (N = 30, in purple; $Oga^{KO}$ experiments were performed simultaneously and first published here [53] with significantly increased bouton counts (p <0.05) without multiple testing correction). * $p_{adj}$ <0.05, ** $p_{adj}$ <0.01, *** $p_{adj}$ <0.001, based on one-way ANOVA with Tukey's multiple comparisons test. A complete list of p-values and summary statistics is provided in S3 Table. (A'- D') Representative NMJs of wandering third instar larvae labeled with anti-discs large 1 (*Dlg*, magenta) and anti-synaptotagmin (*Syt*, green). When appropriate, type 1b synapses are distinguished from other synapses with white arrow. *Scale bar*, 20μm. The quantitative parameter values of the representative images: (A') (+/+ | $sxc^{H537A/+}$): #Boutons (31 | 39), Length (103.7 | 137.8), Area (374.4 | 369.4), Perimeter (245.5 | 306.7) (B') (*elav-Gal4/+* | *UAS-sxc/+* | *Elav-Gal4>UAS-sxc*): #Boutons (45 | 37 | 28), Length (160.0 | 137.5 | 93.3), Area (480.4 | 478.4 | 363.9), Perimeter (407.6 | 290.8 | 243.9) (C') (+/+ | $sxc^{H537A/+}$, *UAS-sxc/+* | $sxc^{H537A/+}$; *elav-Gal4>UAS-sxc*): #Boutons (27 | 34 | 31), Length (107.6 | 144.9 | 114.9), Area (430.6 | 464.7 | 361.7), Perimeter (256.0 | 354.8 | 299.4) (D') (+/+ | $sxc^{H596F/H596F}$ | $Oga^{KO/KO}$ | $sxc^{H596F/H596F}$; $Oga^{KO/KO}$): #Boutons (23 | 43 | 35 | 30), Length (111.5 | 205.6 | 122.9 | 142.5), Area (351.7 | 691.7 | 417.8 | 377.6), Perimeter (245.6 | 544.0 | 274.1 | 318.9).

mutant background ($sxc^{H537A/+}$, *elav-Gal4>UAS-sxc*) (Fig 2C). These results demonstrate a role of *sxc* in the synaptic bouton number and NMJ morphology and indicate that tight control of O-GlcNAcylation is important for normal synaptic development.

The decrease of synaptic length caused by neuronal overexpression of *sxc* in the control but not $sxc^{H537A/+}$ background indicates that strong dysregulation of *sxc* might be required to uncover its function in synaptic growth. Accordingly, we found a significant increase in NMJ length and perimeter in homozygous $sxc^{H537A}$ larvae. The increase in the number of synaptic boutons did not reach significance (S3D Fig), potentially due to greater variability. This effect also did not withstand multiple testing correction in the $sxc^{H537A/+}$; *UAS-sxc/+* larvae that were used as a control in the rescue experiment (Fig 2C). However, there is a quantitatively very consistent increase in bouton number across the three tested H537A mutant conditions (fold changes 1.19, 1.18 and 1.15). We therefore conclude that the bouton phenotype associated with H537A mutation is mild. To validate the synaptic bouton and length/growth phenotypes, we used CRISPR/Cas9 gene editing to generate a stronger catalytic hypomorph, $sxc^{H596F}$ (S1A and S1B Fig). The *in vitro* catalytic activity of $sxc^{H596F}$ was reported to be 3% relative to wild-type OGT activity, less than the reported catalytic activity of $sxc^{H537A}$ (5.6% activity relative to wildtype) [40]. Indeed, we found total O-GlcNAc levels in $sxc^{H596F}$ homozygous embryos to be reduced (S4A Fig). $sxc^{H596F}$ homozygous flies are viable, confirming minimal requirement of endogenous OGT activity for completion of development in *Drosophila* (S4B Fig). We found that NMJs of $sxc^{H596F}$ larvae display a significant increase in synaptic bouton number, NMJ length, area and perimeter (Fig 2D), reflecting a more severe NMJ phenotype and indicating the effect of $sxc^{H537A}$ on synaptic morphology is mild and/or not fully penetrant. We also subjected the $sxc^{H596F}$ flies to light-off jump habituation assay but homozygous as well as heterozygous $sxc^{H596F}$ flies showed impaired jump response (38% and 42% initial jumpers), similar to homozygous $sxc^{H537A}$ and pan-neuronal *sxc* knockdown flies. This precluded the assessment of habituation.

A knockout of *Oga* normalized bouton number and partially also the area and perimeter of the $sxc^{H596F}$ NMJs (Fig 2D). These data show that O-GlcNAcylation controls bouton number and partially also NMJ size.

## Characterization of development and locomotor function of *sxc* mutations associated with Intellectual Disability

Recent studies have reported three hemizygous missense mutations (R248P, A319T, L254F) in human *OGT* in male individuals with ID. The *de novo* R248P mutation was identified by trio

whole exome sequencing in an affected individual with ID and developmental delay [28]. A319T and L254F mutations were identified by X chromosome exome sequencing. The A319T mutation, present in three individuals with severe ID, was inherited from the mother but segregated with an uncharacterized missense mutation in *MED12*, a gene already implicated in ID [25]. The L254F mutation was present in three related individuals with moderate to mild ID [27,29]. These mutations reside in the conserved TPR domain, outside of the catalytic O-GlcNAc transferase domain [25,27–29]. To investigate the functional consequences of these mutations, we introduced the equivalent missense mutations (R313P, A348T, L283F) into the *sxc* gene using CRISPR/Cas9 editing (**S1 Fig**) and generated three novel *sxc* ID alleles *sxc^R313P^*, *sxc^A348T^*, and *sxc^L283F^*.

We first characterized the development of the patient-related mutant *sxc* alleles. We transferred embryos at stage 11–16 to vials with fresh food and counted the number of resulting pupae and adult flies. Homozygous *sxc^R313P^*, *sxc^A348T^*, and *sxc^L283F^* embryos developed normally to adulthood without apparent delay and the percentage of pupae and adults did not statistically differ from the genetic background controls (**S5A Fig**).

We next investigated locomotor phenotypes in adult *sxc^R313P^*, *sxc^A348T^*, and *sxc^L283F^* flies using the island and negative geotaxis assays. In the island assay, flies were thrown onto a white platform surrounded by water, and the number of individuals remaining on the platform was quantified over time. Homozygous *sxc^R313P^*, *sxc^A348T^* and *sxc^L283F^* flies escaped from the platform with similar efficiency as the genetic background control (**S5B Fig**), indicating that their startle response is not affected.

In the negative geotaxis assay, climbing performance of homozygous and heterozygous *sxc^R313P^* flies was significantly slower while homozygous *sxc^A348T^* and *sxc^L283F^* flies showed an average climbing speed similar to the control (**Fig 3A**). We also tested *Drosophila* lines with homozygous H537A (*sxc^H537A^*) or H596F (*sxc^H596F^*) catalytic hypomorph mutations to investigate whether impaired O-GlcNAc transferase activity affects climbing speed in the negative geotaxis assay. The catalytic hypomorphs exhibited similar climbing speed as the control group (**Fig 3A**). This suggests that the deficit in coordinated locomotor behavior in the negative geotaxis assay of *sxc^R313P^* flies is independent of O-GlcNAc transferase activity.

Taken together, similar to catalytic hypomorphs ([51] and **S2B Fig**) the *sxc^R313P^*, *sxc^A348T^*, and *sxc^L283F^* mutants are fully viable and develop normally to adulthood. Neither the reduction of protein O-GlcNAcylation induced by catalytic hypomorph alleles nor the *sxc^A348T^* and *sxc^L283F^* ID alleles cause severe locomotor defects in adult flies. Only the *sxc^R313P^* allele negatively affects climbing performance. This effect appears to be independent of O-GlcNAc transferase activity. However, a contribution of a potential second site mutation affecting another gene that was not eliminated by six generations of backcrossing cannot be formally excluded.

## Patient-related *sxc* mutant alleles do not affect global protein O-GlcNAcylation

We investigated whether the ID-associated alleles in *sxc* affect protein O-GlcNAcylation by subjecting lysates from adult heads of *sxc^R313P^*, *sxc^A348T^*, and *sxc^L283F^* flies to Western blotting. Labeling with anti-O-GlcNAc antibody (RL2) that is able to capture O-GlcNAcylation changes in the catalytic hypomorphs (**S4B Fig** and [40]) indicated that the levels of protein O-GlcNAcylation are not significantly altered in each of the three mutants (**Fig 3B and 3C**). This is in line with normal levels of O-GlcNAcylation in patient-derived fibroblasts and human embryonic stem cell models of the R313P and L283F equivalent mutations [26,28]. O-GlcNAcylation of the human A348T equivalent has not been investigated. The cell models of OGT-CDG mutations show downregulation of Oga, which may compensate for decreased

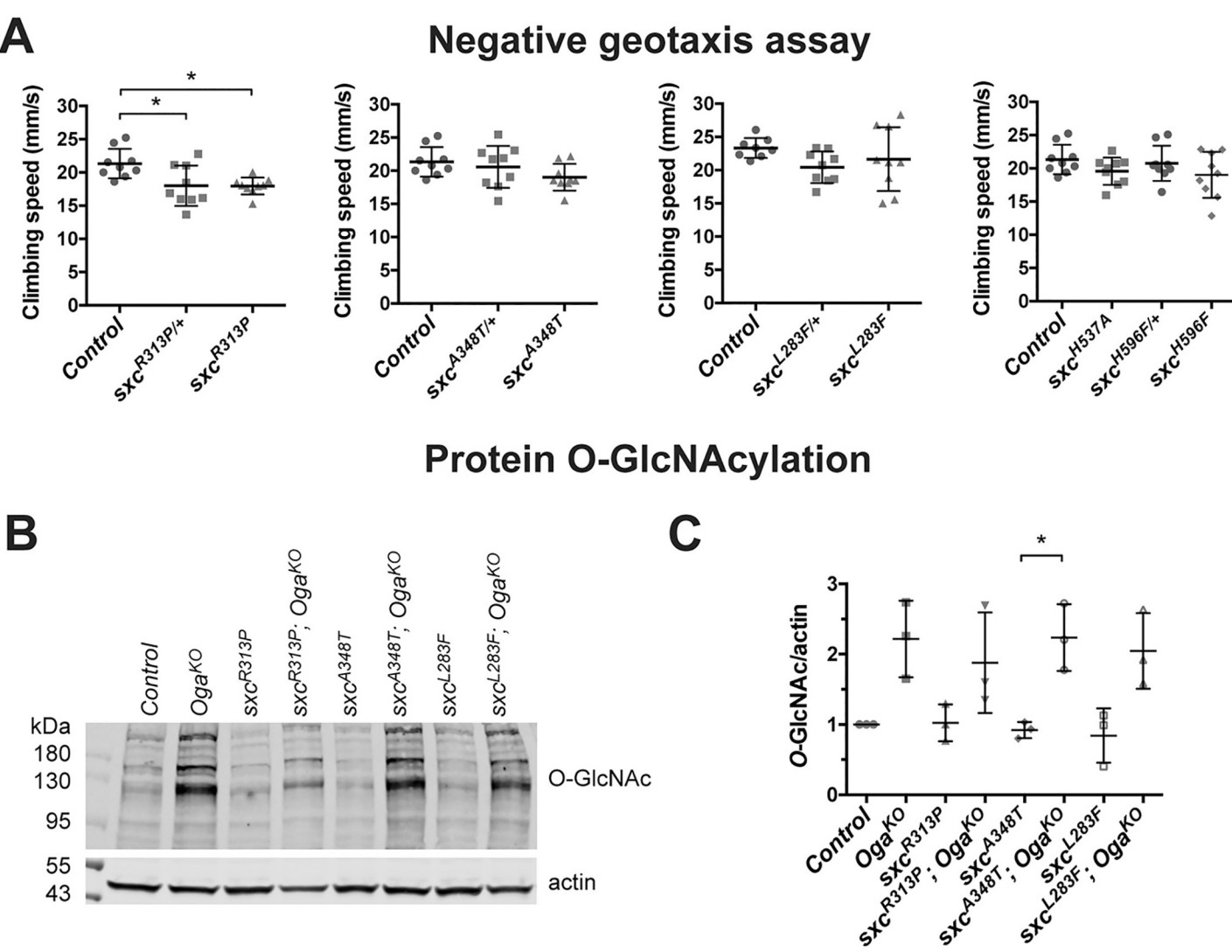

**Fig 3. Locomotor and biochemical characterization of *sxc^R313P^*, *sxc^A348T^* and *sxc^L283F^* flies.** (A) Climbing locomotor behaviour was assessed based on the climbing speed (mm/s) in an automated negative geotaxis assay. The *sxc^R313P/+^* and *sxc^R313P^* (N = 9) flies showed reduced climbing speed compared to background control (N = 9) indicating locomotor dysfunction. *sxc^A348T^*, *sxc^L283F^*, *sxc^H537A^* and *sxc^H596F^* flies (N = 9 for all genotypes) did not show significantly reduced climbing speed. Data presented as mean ± SD. * p_adj < 0.05 based on one-way ANOVA with Tukey's multiple comparisons of mean climbing speed. A complete list of p-values and summary statistics is provided in **S3 Table**. (B) Western blot on head samples from 1–4 days old male adult Drosophila indicate no significant alteration in the level of protein O-GlcNAcylation in *sxc^R313P^*, *sxc^A348T^* and *sxc^L283F^* samples compared to the genetic background controls, while the homozygous *Oga^KO^* allele caused an increase of O-GlcNAcylation. Western blot was probed with a monoclonal anti-O-GlcNAc antibody (RL2). (C) Quantification of O-GlcNAcylated proteins revealed that protein O-GlcNAcylation in *sxc^R313P^*, *sxc^A348T^* and *sxc^L283F^* flies remain at a similar level as in the control samples. Data presented as mean ± SD. * p_adj = 0.035, based on one-way ANOVA with Tukey's multiple comparisons test, n = 3 for all lines. A complete list of p-values and summary statistics is provided in **S3 Table**.

O-GlcNAcylation. Although existence of such regulatory mechanism has not been shown in *Drosophila*, we analyzed the O-GlcNAcylation levels in *Oga* knockout background. Blocking O-GlcNAc hydrolysis in patient-related *sxc* mutant alleles with *Oga^KO^* increased O-GlcNAc levels to the same degree as in *Oga^KO^* samples (**Fig 3B and 3C**). We thus conclude that flies carrying ID-associated *sxc* mutations do not have a grossly affected protein O-GlcNAcylation.

### *sxc^R313P^* and *sxc^A348T^* display defective habituation learning

Because of their role in ID in humans, we also investigated the effect of the novel *sxc^R313P^*, *sxc^A348T^*, and *sxc^L283F^* alleles on habituation learning. We first subjected the *sxc^L283F^* flies to 100

light-off pulses in the habituation assay. Despite sufficient locomotor abilities to perform in the island test and negative geotaxis assays, the initial jump response of the $sxc^{L283F}$ homozygous and heterozygous flies was below the required threshold of 50% therefore deemed non-performers ($sxc^{L283F/L283F}$: 36% initial jumpers, N = 96; $sxc^{L283F/+}$: 47% initial jumpers, N = 96). Insufficient performance at the beginning of the assay thus precluded the assessment of habituation in these flies. We observed the same phenotype also for the $sxc^{R313P}$ homozygous flies (49% initial jumpers, N = 64). The initial response in $sxc^{R313P}$ heterozygous flies was sufficient (67%) and they were not able to suppress their jump response to the repeated light-off stimuli as efficiently as the genetic background control flies (**Fig 4A**), revealing a learning deficit. Flies heterozygous for the $sxc^{A348T}$ allele showed a good initial jump response and habituated similar to the control, while $sxc^{A348T}$ homozygous flies showed a habituation deficit (**Fig 4B**). In summary, deficits in habituation learning were observed for the R313P (heterozygous) and A348T (homozygous) mutations, while evaluation of the L254F homo- and heterozygous as well as R313P homozygous conditions was precluded by a poor initial jump response.

## Blocking O-GlcNAc hydrolysis corrects habituation deficits of $sxc^{R313P}$ and $sxc^{A348T}$

The apparently unaltered levels of O-GlcNAcylation in ID-associated *sxc* mutants (**Fig 3B and 3C**) may suggest that O-GlcNAc-independent mechanisms underlie their cognitive phenotypes. To test this experimentally, we performed the habituation assay in flies carrying $sxc^{R313P}$ allele and either the $Oga^{KO}$ allele or an $Oga$ mutation that specifically blocks its O-GlcNAc hydrolase activity ($Oga^{D133N}$). Notably, heterozygous $Oga^{KO}$ and $Oga^{D133N}$ flies habituated to a similar degree as the genetic background control flies. When introduced into an $sxc^{R313P/+}$ background, $Oga^{KO}$ and $Oga^{D133N}$ alleles fully rescued defects seen in the *sxc* mutants alone (**Fig 4C and 4D**). Similarly, we attempted a rescue of habituation deficient homozygous $sxc^{A348T}$ with homozygous $Oga^{KO}$ and $Oga^{D133N}$ alleles. We have previously shown that these homozygous $Oga$ mutants also exhibit habituation deficits [53]. Strikingly, blocking O-GlcNAc hydrolysis by $Oga^{KO/KO}$ or $Oga^{D133N/D133N}$ in $sxc^{A348T/A348T}$ flies was sufficient to completely rescue habituation deficits of either single mutant condition (**Fig 4E and 4F**). All tested flies show a good initial jump response and lower TTCs in the rescue experiments were not caused by fatigue (**S4 Fig** and **S2 Table**). In summary, despite seemingly normal gross O-GlcNAc levels in the $sxc^{R313P}$ and $sxc^{A348T}$ ID alleles, these genetic experiments provide evidence that their deficits in habituation learning depend on defective OGT enzymatic activities.

## $sxc^{R313P}$ and $sxc^{A348T}$ and $sxc^{L283F}$ show deficits in synaptic morphology

To determine whether synaptic phenotypes seen in larvae with hypomorphic catalytic domain mutations are recapitulated in larvae with patient mutations we assessed synaptic morphology at the NMJs of homozygous $sxc^{R313P}$ and $sxc^{A348T}$ and $sxc^{L283F}$ larvae. We found that the NMJs of larvae carrying any of the three patient-related *sxc* mutant alleles display a significant increase in synaptic bouton number and $sxc^{R313P}$ larvae also display significantly increased length and perimeter of the NMJ. NMJ length is also increased in the $sxc^{A348T}$ and $sxc^{L283F}$ larvae albeit not significantly (**Fig 5A**). In addition, $sxc^{R313P}$ and $sxc^{L283F}$ larvae show an increased number of NMJ branches (**S5C Fig**). Overall, the NMJ morphology phenotypes of the patient-related *sxc* mutant alleles resemble those of the catalytic hypomorphs $sxc^{H537A/+}$ (**Fig 2A**) and $sxc^{H596F/H596F}$ (**Fig 2D**). This is in line with the observation that the patient-related $sxc^{R313P}$ and $sxc^{A348T}$ alleles affect O-GlcNAc transferase activity, which is indispensable for habituation

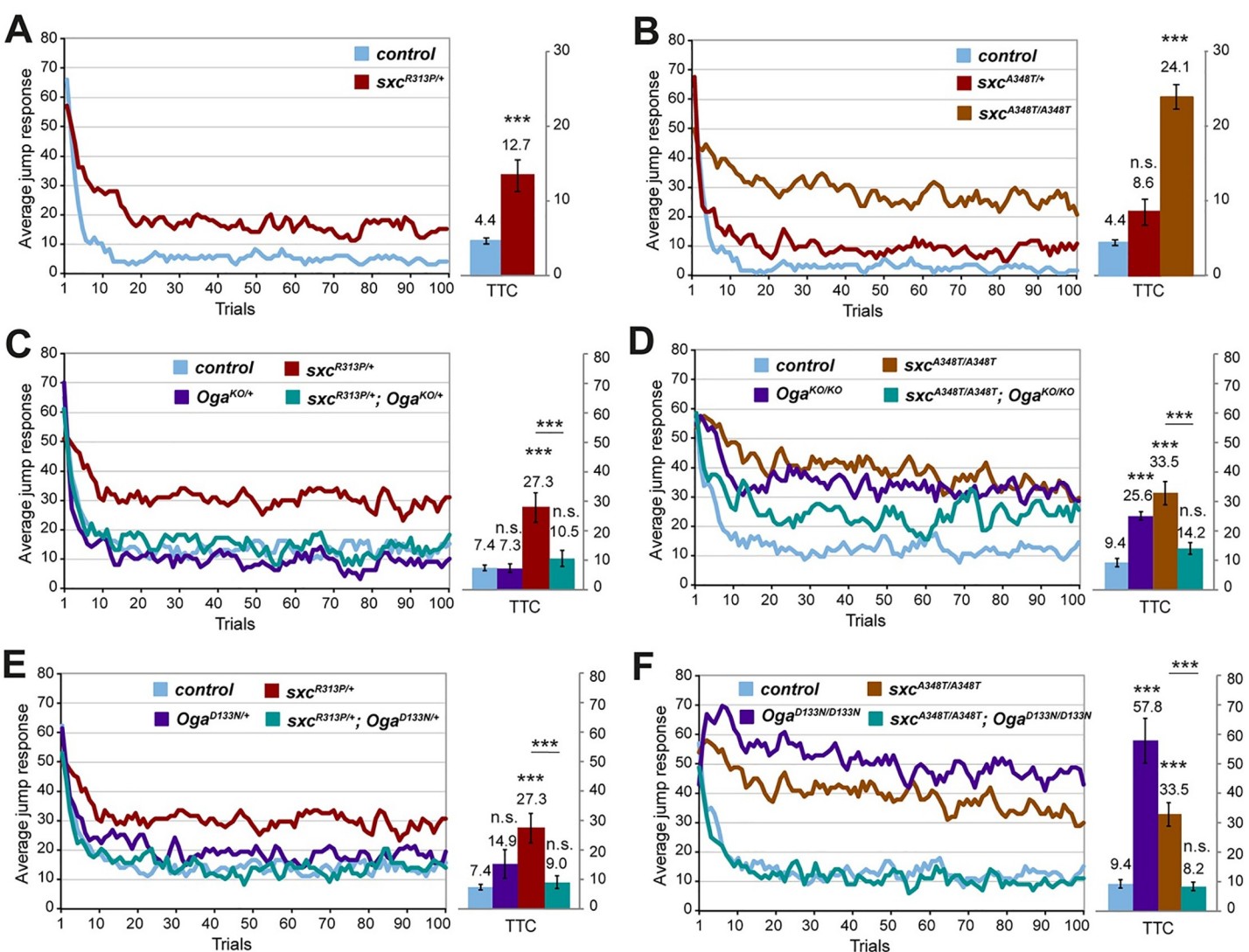

**Fig 4. Assessment of *sxc^R313P* and *sxc^A348T* flies in habituation learning.** Jump responses were induced by 100 light-off pulses with 1 s interval between pulses. The jump response represents the % of jumping flies in each light-off trial. The mean number of trials that flies needed to reach the no-jump criterion (Trials To Criterion, TTC) ± SEM is also shown. (A) Deficient habituation of *sxc^R313P/+* flies (N = 73, $p_{adj}$ = 6.18x10^-6, in red) compared to their respective genetic background controls (*control*, N = 65, in blue). (B) Deficient habituation of *sxc^A348T/A348T* flies (N = 76, $p_{adj}$ = 2.1x10^-14, in brown) and no significant habituation deficit of *sxc^A348T/+* flies (N = 72, $p_{adj}$ = 0.095, in red) compared to the genetic background control (*control*, N = 65, in blue). (C) Deficient habituation of *sxc^R313P/+* flies (N = 81, $p_{adj}$ = 2.63x10^-6, in red) is restored in *sxc^R313P/+*; *Oga^KO/+* flies (N = 53, $p_{adj}$ = 4.89x10^-5, in cyan). (D) Deficient habituation of *sxc^A348T/A348T* flies (N = 79, $p_{adj}$ = 8.84x10^-10, in brown) is restored in *sxc^A348T/A348T*; *Oga^KO/KO* flies (N = 62, $p_{adj}$ = 1.07x10^-4, in cyan). (E) Deficient habituation of *sxc^R313P/+* flies (N = 81, $p_{adj}$ = 2.63x10^-6, in red) is restored in *sxc^R313P/+*; *Oga^D133N/+* flies (N = 64, $p_{adj}$ = 9.09x10^-6, in cyan). (F) Deficient habituation of *sxc^A348T/A348T* flies (N = 79, $p_{adj}$ = 8.84x10^-10, in brown) is restored in *sxc^A348T/A348T*; *Oga^D133N/D133N* flies (N = 56, $p_{adj}$ = 3.74x10^-7, in cyan). *** $p_{adj}$<0.001, n.s. not significant, based on lm analysis with Bonferroni-Holm correction for multiple comparisons. A complete list of p-values and summary statistics is provided in S3 Table.

(Fig 4C and 4F). The shared NMJ phenotype signature between the catalytic and patient-related mutants is the increase of synaptic bouton number. Because R313P and L283F mutations also significantly affect other NMJ parameters, we conclude that they are stronger/more detrimental than A348T mutation. This is in line with the observed effect on the jump performance in the light-off jump habituation assay (*sxc^L283F/L283F*, *sxc^L283F/+* and *sxc^R313P/R313P* non-performers).

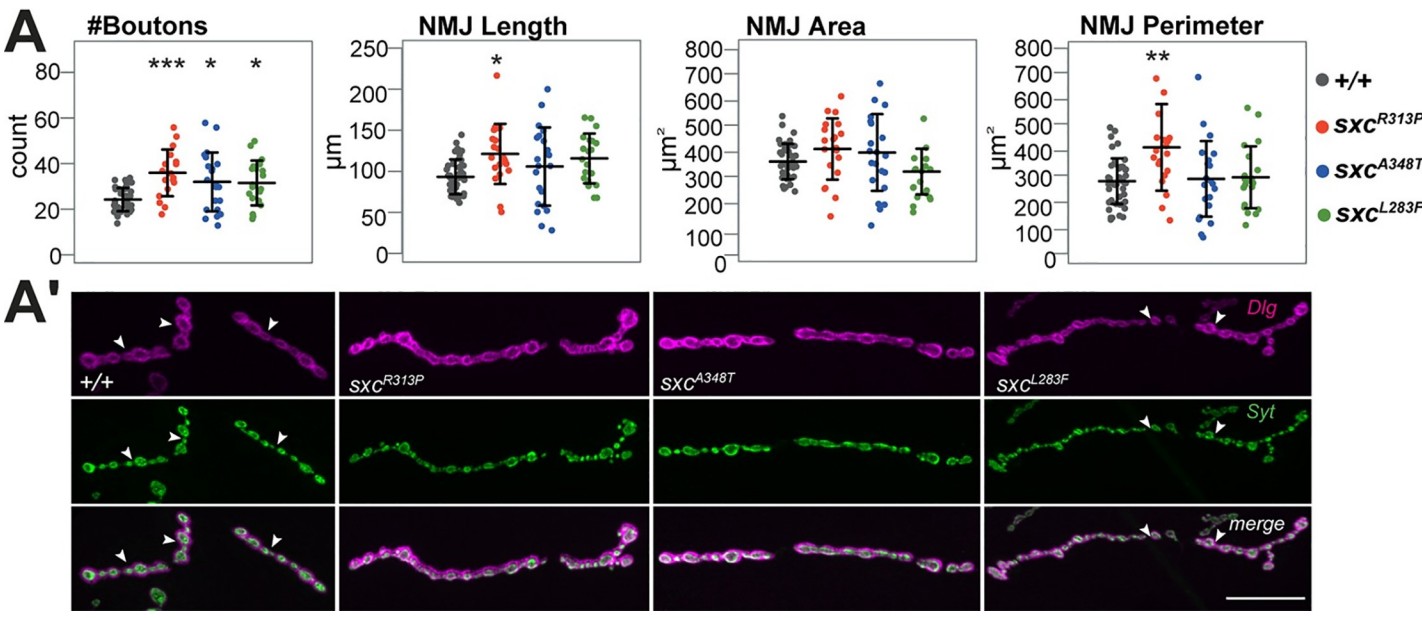

**Fig 5. Synaptic morphology of *sxc^R313P^*, *sxc^A348T^* and *sxc^L283F^*.** Data presented as individual data points with mean ± SD. (A) NMJs on muscle 4 of *sxc^R313P^*, *sxc^A348T^* and *sxc^L283F^* larvae have a significantly higher number of synaptic boutons (*sxc^R313P^*: N = 21, p = 6.8x10^-5, in red; *sxc^A348T^*: N = 21, p = 0.011, in blue; *sxc^L283F^*: N = 20, p = 0.0225) compared to the control (+/+, N = 42, in grey). *sxc^R313P^* larvae have also significantly higher NMJ length (p = 0.0125) and perimeter (p = 0.001).). * $p_{adj}$ <0.05, ** $p_{adj}$ <0.01, *** $p_{adj}$ <0.001, based on one-way ANOVA with Tukey's multiple comparisons test. A complete list of p-values and summary statistics is provided in **S3 Table**. (A') Representative NMJs of wandering third instar larvae labeled with anti-discs large 1 (*Dlg*, magenta) and anti-synaptotagmin (*Syt*, green). When appropriate, type 1b synapses are distinguished from other synapses with white arrow. *Scale bar*, 20μm. The quantitative parameter values of the representative images (+/+ | *sxc^R313P^* | *sxc^A348T^* | *sxc^L283F^*): #Boutons (20 | 36 | 29 | 37), Length (84.5 | 113.8 | 107.4 | 128), Area (399 | 447.9 | 360.9 | 349.9), Perimeter (254.2 | 329.5 | 313.8 | 302.4).

### The *Drosophila* O-GlcNAc proteome is enriched in genes with function in neuronal development, learning & memory, and human ID gene orthologs

ID-associated mutations in *sxc* do not globally reduce protein O-GlcNAcylation yet blocking O-GlcNAc hydrolysis can completely restore the learning deficits in light-off jump habituation. We hypothesized that altered O-GlcNAcylation of specific *sxc* substrates is responsible for the habituation deficits. We therefore attempted to predict candidate substrates by exploration of the *Drosophila* O-GlcNAc proteome, as previously determined through enrichment with a catalytically inactive bacterial O-GlcNAcase [60]. We first performed an enrichment analysis of neuronal and cognitive phenotypes (as annotated in Flybase, see Materials and Methods) among the *Drosophila* O-GlcNAc substrates (encoded by in total 2293 genes) and found that they are significantly enriched in phenotype categories learning defective (Enrichment = 1.5, $p_{adj}$ = 0.032), memory defective (Enrichment = 1.5, $p_{adj}$ = 0.016), neurophysiology defective (Enrichment, 1.5, $p_{adj}$ = 3.2x10^-5) and neuroanatomy defective (Enrichment = 1.9, p = 5.27x10^-35) (genes listed in **S4 Table**), supporting the importance of O-GlcNAcylation for neuronal development and cognitive function. We also found that human orthologs of 269 genes from the O-GlcNAc proteome are proven or candidate monogenic causes of ID (Enrichment = 1.7, $p_{adj}$ = 1.01x10^-16). When restricting this analysis to proteins with high-confidence mapped O-GlcNAc sites (in total 43) [60], we found orthologs of nine O-GlcNAcylated proteins to be implicated in Intellectual Disability, again representing a significant enrichment (Enrichment = 3.4, $p_{adj}$ = 0.01). These orthologs are: Atpalpha (human ATP1A2), Gug (human ATN1), Hcf (human HCF1), LanA (human LAMA2), mop (human PTPN23), NAChRalpha6 (human CHRNA7), Ndg (human NID1), Nup62 (human NUP62) and Sas-4 (human CENPJ).

They represent potential downstream effectors of *sxc* that may control habituation learning in the wild-type condition and may contribute to habituation deficits in catalytic and ID-associated *sxc* mutant conditions. Further analysis will be required to answer the question whether impaired O-GlcNAc transferase activity towards one or more of these targets is responsible for habituation deficits that are associated with OGT-CDG mutations.

## Discussion

### O-GlcNAcylation is important for habituation and for neuronal development in *Drosophila*

Habituation, the brain's response to repetition, is a core element of higher cognitive functions [44–46]. Filtering out irrelevant familiar stimuli as a result of habituation allows to focus the cognitive resources on relevant sensory input. Abnormal habituation was observed in a number of neurodevelopmental disorders, including ID and Autism [47] and characterizes > 100 *Drosophila* models of ID [48,49]. To address the role of OGT and its O-GlcNAc transferase activity in this cognition-relevant process, we investigated heterozygous $sxc^{H537A/+}$ flies [51] in light-off jump habituation. We found that they were not able to suppress their escape behavior as a result of deficient habituation learning (**Fig 1A**). This finding is in line with the recently published habituation deficit of the complete knock-out of *OGT* ortholog in *C. elegans* [61] and shows that altering O-GlcNAc transferase activity is sufficient to induce this deficit. We thus demonstrate the importance of O-GlcNAcylation in habituation learning.

Proper development and maintenance of synapses is an important aspect of neuronal function and cognition. The synaptic connection between motor neurons and muscle cells, termed the neuromuscular junction (NMJ), represents an excellent model system to study the molecular mechanisms of synaptic development in *Drosophila* [62]. Because NMJ defects were found in several *Drosophila* disease models with defective habituation [48,54–56], we investigated the synaptic architecture of the $sxc^{H537A/+}$ larvae. We found that NMJs of the $sxc^{H537A/+}$ larvae are characterized by an increased number of synaptic boutons, recognizable structures that contain the synaptic vesicles (**Fig 2A**). Larvae with a stronger homozygous catalytic mutation, $sxc^{H596F/H596F}$, also show an increase in NMJ length, area, and perimeter. We conclude that O-GlcNAcylation is important for control of synaptic size and synaptic bouton number.

### Appropriate O-GlcNAc cycling is required for habituation learning and maintenance of the synaptic size

We recently showed that increased protein O-GlcNAcylation in homozygous *Oga* knockout flies causes a habituation deficit [53]. Here we show that heterozygous *Oga* knockout can restore the habituation deficit of $sxc^{H537A/+}$ flies (**Fig 1B**). This indicates that habituation learning depends on O-GlcNAc cycling. Because the loss of one *Oga* allele does not significantly affect total O-GlcNAc levels [53], we presume that subtle changes in O-GlcNAcylation dynamics rather than gross loss of O-GlcNAc transferase activity inhibits habituation learning.

It is known that postsynaptic expression of OGT in excitatory synapses is important for synapse maturity in mammals [24]. Here we show that presynaptic O-GlcNAc transferase also has role in synapse growth. At the NMJ, the synapses of larvae with neuronal overexpression of *sxc* are shorter, and the number of synaptic boutons is decreased (**Fig 2B**). Both length and bouton number are normalized when *sxc* is overexpressed in neurons of the $sxc^{H537A/+}$ larvae (**Fig 2C**). This phenotype was not observed in *Oga* knockout larvae with increased O-GlcNAcylation. Knockout of *Oga* can correct the increased bouton number in larvae with $sxc^{H596F}$ mutation but not the NMJ size (**Fig 2D**).

Our data suggest that the NMJ defects associated with decreased O-GlcNAc transferase function are of neuronal origin and that O-GlcNAcylation controls the number of synaptic boutons and partially also synaptic size. Absence of synaptic size defects in *Oga* knockout larvae and failure of *Oga*$^{KO}$ to rescue the NMJ size defects caused by decreased O-GlcNAcylation indicates that other, non-catalytic O-GlcNAc transferase functions may be involved in the control of synaptic size. Levine et al. recently demonstrated that non-catalytic activities of OGT are necessary for its function in some cellular processes, such as proliferation [38].

### *Drosophila* NMJ as a model for the O-GlcNAc-related synaptopathy

The *sxc* catalytic hypomorph mutations (*sxc*$^{H537A}$, *sxc*$^{H596F}$) as well as the OGT-CDG-patient equivalent mutations (R284P, A319T, L254F) that we introduced with the CRISPR/Cas-9 gene-editing technology in the *Drosophila sxc* gene (*sxc*$^{R313P}$, *sxc*$^{A348T}$, *sxc*$^{L283F}$), lead to an increase in the number of synaptic boutons, and in some cases also to an increase in synaptic size. NMJ size and the number of synaptic boutons in our model is determined by the level of *sxc* activity. Dependence of these parameters on gene activity/dosage was previously established in *Fmr1* (the *Drosophila* model of Fragile X Syndrome) [59] and other *Drosophila* models of neurodevelopmental or neurological disorders, including *Prosap/SHANK* mutants (modelling Phelan-McDermid Syndrome caused by mutations in *SHANK3*, characterized by ID and ASD), *Neuroligin 4* (ID and ASD caused by mutations in *NLGN4*), *VAP33* (model of Amyotrophic Lateral Sclerosis caused by mutations in *VAP-33A*) and *highwire* (potential therapeutical target in traumatic brain injury) [63–66]. The synaptic phenotypes associated with impaired *sxc* catalytic activity may be linked to increased microtubule polymerization, since it has been shown that O-GlcNAcylation of tubulin negatively regulates microtubule polymerization and neurite outgrowth in mammalian cell lines [67] and *Fmr1* and *VAP-33A* control synaptic growth and bouton expansion through presynaptic organization of microtubules [59,66].

Increased number of synaptic boutons has been also associated with increased excitability at the NMJ [68–70] although not consistently [70–72]. An interesting future direction could involve electrophysiological assessment of NMJ activity to determine whether O-GlcNAc cycling and the patient-related *sxc* mutations go beyond determining synapse development and affect synapse excitability and/or plasticity. However, these investigations would need to test various aspects of physiology and would still leave the impact of O-GlcNAc on cognition undetermined. For this reason, we assessed habituation as a highly cognition-relevant parameter.

### Mutations implicated in OGT-CDG affect habituation via modulation of O-GlcNAc transferase activity

We assessed the effect of OGT-CDG missense mutations on habituation. We found that *sxc*$^{R313P}$ and *sxc*$^{A348T}$ inhibit habituation in the light-off jump habituation assay (**Fig 4A and 4B**). *sxc*$^{L283F}$ could not be investigated as these mutants displayed a non-performer phenotype in the light-off jump response. While the full spectrum of ID-related phenotypes in an individual with R284P mutation has been attributed to *OGT*, the A319T mutation segregates with an uncharacterized missense mutation in another gene implicated in ID, *MED12* (G1974H) [25]. It was not known which of the mutations is responsible for ID in the affected individuals. We provide evidence that the *Drosophila* equivalent of the A319T mutation in the TPR domain of OGT causes a cognitive deficit and support a causal role of A319T in OGT-CDG.

Consistent with no detectable O-GlcNAc changes in patient samples and cellular models of the non-catalytic OGT mutations [26,28], no appreciable reduction in protein O-GlcNAcylation was observed in *sxc*$^{R313P}$ and *sxc*$^{A348T}$ flies. However, habituation learning was restored by increasing O-GlcNAcylation through blocking Oga activity (**Fig 4C–4F**). This argues that the

mechanism by which $sxc^{R313P}$ and $sxc^{A348T}$ inhibit habituation is defective O-GlcNAc transferase activity, paralleling impaired O-GlcNAc transferase activity and significant reduction of protein O-GlcNAcylation demonstrated in the catalytic OGT-CDG mutations [30,31]. It is worth noticing that we have previously shown that mutations in *Oga* also cause habituation deficits [53]. Our finding that genetic combination of loss of OGA with loss of OGT activity rescues the cognitive readout argues that OGA inhibition using available inhibitors may represent a viable treatment strategy. The R284P and A319T reside in the TPR domain (**S1 Fig**), which is responsible for recognition and binding of OGT substrates [38,73,74]. All OGT-CDG mutations investigated in this study were shown to impair the substrate interaction properties and the glycosyltransferase kinetics [27,36]. The observed habituation deficits may thus be caused by impaired O-GlcNAcylation dynamics towards a specific set of substrates that cannot be captured by standard O-GlcNAc detection assays. Identification of these substrates may pinpoint the underlying defective mechanisms and additional treatment targets.

## Potential downstream effectors of O-GlcNAcylation in cognition and cognition-relevant processes

Our explorative analysis found that of 43 established O-GlcNAcylated proteins [60], nine are orthologs of human proteins implicated in ID: ATP1A2, ATN1, HCF1, LAMA2, PTPN23, CHRNA7, NID1, NUP62 and CENPJ. These proteins represent potential downstream effectors and can be investigated in future studies. Particularly the transcriptional co-regulator HCF1 (Host Cell Factor 1) emerges as a top candidate. In mammals, OGT mediates glycosylation and subsequent cleavage of HCF1, which is essential for its maturation [75]. Recombinant OGT with an R284P amino acid substitution is defective in HCF1 glycosylation [28] and HCF1 processing was shown to be completely abrogated by a catalytic OGT-CDG mutation [30].

*Drosophila* sxc is a member of the polycomb group (PcG), a conserved set of chromatin and transcriptional modifiers that initially have been identified by phenotypic similarity of their mutant phenotypes: homeotic transformations. They are required for maintenance of transcriptional repression (of non-lineage genes) during embryonic development and cell proliferation [76,77]. Missing O-GlcNAcylation of PcG component Polyhomeotic (Ph) is responsible for misexpression of *HOX* genes and homeotic transformations in *sxc* null mutants [78]. A recent study has shown that chromatin redistribution induced by interaction between sxc and PcG member Polycomb like (Pcl) controls plasticity of sensory taste neurons [79]. It is not known whether the regulation of PcG activity by sxc/OGT is important for cognitive function, but it can be noted that a series of PcG genes are associated with ID [80,81], and some of them are subject to regulation by OGT in the context of development or cancer. These include *PHC1* –human ortholog of *Drosophila Ph* [82], *RING1B* (*Drosophila Sce*) [83,84], *EZH2* (*Drosophila E(z)*) [85–87], *YY1* (*Drosophila pho*) [88,89] and *ASXL1* (*Drosophila Asx*) [90,91]. In addition, OGT regulates expression of PcG genes by O-GlcNAcylation of PcG transcriptional regulators, for example *ATN1* (*Drosophila Gug*), which was identified in the embryonic O-GlcNAc proteome [60]. The encoded proteins represent interesting candidate targets that may link cognitive deficits of OGT-CDG mutations to PcG function.

We propose that in depth clinical phenotyping of patients with mutations in OGT and the above listed genes may give additional hints to the most crucial downstream targets of OGT-mediated O-GlcNAcylation.

In summary, we show that OGT-CDG mutations in the TPR domain negatively affect habituation learning in *Drosophila* via reduced protein O-GlcNAcylation. The data support a causal role of A319T in OGT-CDG and demonstrate that *Drosophila* habituation can be used

to analyze the contribution of OGT mutations to cognitive deficits. This important aspect of ID has to date not been addressed for any of the OGT-CDG mutations. Moreover, our genetic approach points to a key role of O-GlcNAc transferase activity in ID-associated cognitive deficits and identifies blocking O-GlcNAc hydrolysis as a treatment strategy that can ameliorate cognitive deficits in OGT-CDG patients. Thanks to its high-throughput compatibility, the light-off jump habituation assay can be used with high efficiency for future identification of the downstream effectors and novel therapeutic targets for OGT-CDG.

## Materials and methods

### Cloning of the guide RNA and repair template DNA vectors for *Drosophila* CRISPR/Cas9 editing

Novel mutant *Drosophila* lines, $sxc^{H596F}$, $sxc^{R313P}$, $sxc^{A348T}$, and $sxc^{L283F}$, were generated via CRISPR/Cas9 gene editing, following a previously described protocol [51]. Briefly, guide RNA sites were selected using an online tool (crispr.mit.edu) and the annealing primer pairs with appropriate overhangs for *Bpi*I restriction digestion were cloned into pCFD3-dU63gRNA plasmid [92]. Vectors coding for repair template DNA of roughly 2 kb were generated from *Drosophila* Schneider 2 cell genomic DNA by PCR using GoTaq G2 Polymerase (Promega) and primer pairs appropriate for the desired region (S1 Table). The PCR products were digested with *Bpi*I and inserted into the pGEX6P1 plasmid. The intended mutation, as well as silent mutations required to remove the gRNA sequence (S1 Fig), were incorporated by either site-directed mutagenesis (H596F) using the QuikChange kit (Stratagene) or restriction-free cloning (R313P, A348T and L283F) [93]. The four sets of mutations–H596F, L283F, R313P, and A348T removed restriction sites for *Hinf*I, *Bfm*I, *Mnl*I and *Bsq*I, respectively. DNA products of cloning and mutagenesis were confirmed by sequencing. All primer sequences are listed in S1 Table.

### Generation of $sxc^{H596F}$, $sxc^{R313P}$, $sxc^{A348T}$, and $sxc^{L283F}$ *Drosophila* lines

Vasa::Cas9 *Drosophila* embryos (strain #51323 from Bloomington *Drosophila* Stock Center; bdsc.indiana.edu) were injected with a mixture of CRISPR/Cas9 reagents, 100 ng/μl guide RNA plasmid and 300 ng/μl repair template DNA vector (University of Cambridge fly facility). Injected male flies were crossed with an in-house Sp/CyO balancer stock for two generations, allowing for the elimination of the vasa::Cas9 carrying X chromosome. Candidate F1 males were genotyped exploiting restriction fragment length polymorphism. All lines were validated by sequencing the region approximately 250 base pairs upstream and downstream of the mutations and sequencing the areas outside the repair templates. In addition, all of the predicted off-target sites were PCR-amplified and checked for the presence of any lesions compared with the genomic DNA from the *BL51323* line. None of the predicted off-target sites were found to have mutations. To eliminate any other potential off-target mutations introduced during CRISPR, all lines were backcrossed into the $w^{1118}$ control genetic background for six generations.

### Restriction fragment length polymorphism assay

To assess and confirm the presence of the H596F, L283F, R313P, and A348T mutations in the *sxc* gene, DNA of candidate individual adult flies was extracted using 10–50 μl of DNA extraction buffer containing 10 mM Tris-HCl pH 8, 1 mM EDTA, 25 mM NaCl and 200 μg/ml freshly added Proteinase K (Roche). The solution was subsequently incubated at 37˚C for 30 min, followed by inactivation of Proteinase K at 95˚C for 3 min, and centrifuged briefly. 1 μl of the crude DNA extract was used per 25 μl PCR reaction with the relevant diagnostic primers, using a 2x GoTaq G2 Green premix (Promega). 5 μl of the PCR products were used for

restriction fragment length polymorphism assay with the appropriate enzymes, followed by agarose gel electrophoresis of the digested products. Reactions which showed the presence of an undigested full-length PCR product resistant to the expected restriction enzyme cleavage indicated CRISPR/Cas9 gene editing event and were sequenced. Precise incorporation of the repair template into the right position of the genome was confirmed by sequencing a second round of PCR products obtained from potential homozygous CRISPR mutants with mixed diagnostic and line-check primer pairs. Primer sequences are listed in **S1 Table**.

## Fly stocks and maintenance

*Drosophila* stocks and experimental crosses were reared on a standard *Drosophila* diet (sugar/cornmeal/yeast). An RNAi strain to knockdown *sxc* (*#18610*) and a genetic background control strain (*#60000*) were obtained from the Vienna *Drosophila* Resource Center (VDRC; www.vdrc.at). In-house *sxc*$^{H537A}$ [51] and *UAS-sxc* [40] strains, and the generated *sxc*$^{H596F}$, *sxc*$^{R313P}$, *sxc*$^{A348T}$, and *sxc*$^{L283F}$ strains, were crossed into the VDRC *w*$^{1118}$ control genetic background (*#60000*) for six generations. The *sxc*$^{H537A}$/*CyO; UAS-sxc* strain was assembled using the isogenic strains. *Oga*$^{KO}$ and *Oga*$^{D133N}$ lines were also crossed to this background as described earlier [53]. *#60000* was used as isogenic control for the mutant alleles. In the neuromuscular junction analysis of sxc$^{R313P}$, sxc$^{A348T}$ and sxc$^{L283F}$ alleles, the control flies were derived by crossing the flies from the stock used for microinjection (Bloomington Stock: BL51323) and same crossing scheme as that used to derive the sxc$^{R313P}$, sxc$^{A348T}$ and sxc$^{L283F}$ homozygotes and eliminate the Cas9 transgene were used. To induce neuronal knockdown and overexpression, a *w*$^{1118}$*; 2xGMR-wIR; elav-Gal4, UAS-Dicer-2* driver strain was used. This strain contains a double insertion of an RNAi construct targeting the gene *white* specifically in the *Drosophila* eye (*2xGMR-wIR*) to suppress pigmentation, as required for an efficient light-off jump response [54,55]. Progeny of the crosses between the driver, RNAi/*UAS-sxc* and *#60000* strain were used as controls for knockdown and overexpression experiments. All crosses were raised at 25˚C, 70% humidity, and a 12:12h light-dark cycle.

## Western blotting

Protein lysates for Western blotting were prepared from adult male (1–4 days old) fly head samples or 0–16 h embryo collection and snap frozen in liquid nitrogen. Samples were homogenized in lysis buffer containing 2x NuPAGE LDS Sample Buffer, 50 mM Tris- HCl (pH 8.0), 150 mM NaCl, 4 mM sodium pyrophosphate, 1 mM EDTA, 1 mM benzamidine, 0.2 mM PMSF, 5 μM leupeptin, and 1% 2-mercaptoethanol. Crude lysates were then incubated for 5 min at 95˚C, centrifuged at 13000 rpm for 10 min, and supernatants were collected. Pierce 660 nm protein assay supplemented with Ionic Detergent Compatibility Reagent (Thermo Scientific) was used to determine protein concentration. 20–30 μg of protein samples were separated on RunBlue 4–12% gradient gels (Expedeon) using MOPS running buffer, before being transferred onto nitrocellulose membranes. Western blot analysis was carried out with anti-O-GlcNAc (RL2, Abcam, 1:1000) and anti-actin (Sigma, 1:5000) antibodies. Membranes were incubated overnight with selected primary antibodies in 5% BSA at 4˚C. Blots were visualized via Li-Cor infrared imaging with Li-Cor secondary antibodies (1:10000) Signal intensities were quantified using ImageStudioLite software. Significance was calculated using one-way ANOVA with Tukey's multiple comparisons test ($p_{adj}$).

## Developmental survival

Stage 11–16 embryos (25 embryos per vial, 100 per genotype per experiment, n = 3) were cultured at 25˚C and assessed for lethality by counting the number of pupae and adults derived.

Significance was calculated using Student's t-test with Holm-Sidak's correction for multiple comparisons when appropriate.

### Light-off jump habituation

The light-off jump reflex habituation assay was performed as previously described [49,94]. Briefly, 3- to 7-day-old individual male flies were subjected to the light-off jump reflex habituation paradigm in two independent 16-chamber light-off jump habituation systems. Male progeny of the appropriate control genetic background was tested simultaneously on all experimental days. Flies were transferred to the testing chambers without anesthesia. After 5 min adaptation, a total of 32 flies (16 flies/system) were simultaneously exposed to a series of 100 short (15 ms) light-off pulses with 1 s interval. The noise amplitude of wing vibration following every jump response was recorded for 500 ms after the start of each light-off pulse. A carefully chosen automatic threshold was applied to filter out background noise and distinguish it from jump responses. Data were collected by a custom-made Labview Software (National Instruments). Initial jump responses to light-off pulse decreased with the increasing number of trials and flies were considered habituated when they failed to jump in five consecutive trials (no-jump criterion). Habituation was quantified as the number of trials required to reach the no-jump criterion (Trials To Criterion (TTC)). All experiments were done in triplicates (N = 96 flies). Main effects of genotype on log-transformed TTC values were tested using a linear model regression analysis (lm) in the R statistical software (R version 3.0.0 (2013-04-03)) [95] and corrected for the effects of testing day and system. Bonferroni-Holm correction for multiple testing [96] was used to calculate adjusted p-values ($p_{adj}$).

### Fatigue assay

Each genotype that was tested in light-off jump habituation was subsequently subjected to fatigue assay. The fatigue assay was used to evaluate whether the lower TTCs in the rescue experiments were not a result of increased fatigue rather than improved habituation/non-associative learning. The assay was performed as previously described [49]. The interval between light-off pulses was increased to 5 seconds, an intertrial interval that is sufficiently long to prevent habituation. The light-off pulse was repeated 50 times. Fatigue was concluded when log-transformed TTC values of the rescue were significantly smaller than log-transformed TTC values of the control (based on lm analysis and Bonferroni-Holm correction; $p_{adj} < 0.05$).

### Analysis of *Drosophila* neuromuscular junction

Wandering male L3 larvae were dissected with an open book preparation [97], and fixed in 3.7% paraformaldehyde for 30 minutes. Larvae were stained overnight at 4˚C with the primary antibodies against synaptic markers Discs large (anti-dlg1, mouse, 1:25, Developmental Studies Hybridoma Bank) and synaptotagmin (anti-Syt, rabbit, 1:2000, kindly provided by H. Bellen). Secondary antibodies anti-mouse Alexa 488 and anti-rabbit Alexa 568 (Invitrogen) were applied for 2 hours at room temperature (1:500). Projections of type 1b neuromuscular junctions (NMJs) at muscle 4 from abdominal segments A2-A4 were assessed. Individual synapses were imaged with a Zeiss Axio Imager Z2 microscope with Apotome and quantified using in-house developed Fiji-compatible macros [98,99]. Anti-dlg1 (4F3 anti-discs large, DSHB, 1:25) labeling was used to analyze NMJ area, length, number of branches and branching points. Anti-Syt (kind gift of Hugo Bellen, 1:2000) labeling was used to analyze the number of synaptic boutons. Secondary antibodies goat anti-mouse Alexa Fluor 488 (1:200) and goat anti-rabbit Alexa Fluor 568 from Life Technologies were used for visualization. Parameters with a normal distribution (area, length, number of boutons) were compared between the mutants and controls with one-way

ANOVA ($p$) and Tukey's test for multiple comparisons ($p_{adj}$). Parameters without normal distribution (number of branches and branching points) were compared with non-parametric Wilcoxon test ($p$, single comparisons) and Kruskal-Wallis test with Wilcoxon pairwise test for multiple comparisons ($p_{adj}$) in the R statistical software (R version 3.0.0 (2013-04-03)) [95].

### Island assay

Locomotor behaviour of 3–6 days old male flies was assessed with the island assay as described previously [100,101]. Each trial was performed using 15 flies. 3–4 repeats were carried out on each test day, and data was collected on 3 consecutive days. In total, data from 11–16 trials were collected per genotype. The percentage of flies on the island platform over time was plotted and area under curve (AUC) was determined for each run. Groups were compared using one-way ANOVA with Holm-Sidak's multiple comparisons ($p_{adj}$) of means for AUC.

### Negative geotaxis test

The negative geotaxis assay was performed as described previously [102]. The climbing ability of 3–6 days old male flies was evaluated on groups of 10 animals. Prior to the measurement, flies were transferred into 150 x 16 mm transparent plastic test tubes without anesthesia. Test tubes were secured into a frame that allowed for monitoring of climbing behavior of up to 10 vials at once. Upon release, the frame is dropped from a fixed height onto a mouse pad, thereby tapping the flies to the bottom of the tubes. The climbing assay was repeated 4 times for each loaded frame providing data from 4 runs. The experiment was video-recorded with a Nikon D3100 DSLR camera. ImageJ/FIJI software was used to analyse the resulting recordings. First, images were converted to an 8-bit grey scale TIFF image sequence (10 frames per second) file format. Background-subtraction and filtering were then applied, and the image pixel values were made binary. The MTrack3 plug-in was used for tracking of flies. Mean climbing speed (mm/s) was quantified for each genotype in 2nd, 3rd and 4th runs, between 17–89 data points were collected per run. Groups were compared using one-way ANOVA with Tukey's multiple comparisons ($p_{adj}$) of means on mean climbing speed values calculated for each run.

### Enrichment analysis

The O-GlcNAc proteome data was extracted from Selvan et al. (Supplementary dataset 3) [60]. Phenotype annotations of *Drosophila* gene alleles were extracted from Flybase (Flybase.org, downloaded in April 2016). Human genes implicated in Intellectual disability (ID + ID candidate genes) were extracted from sysid database (https://sysid.cmbi.umcn.nl/, downloaded in April 2016). Enrichment was calculated as follows: (a/b)/((c-a)/(d-b)), whereby a = genes in O-GlcNAc proteome and associated with the phenotype term/human ID gene orthologs, b = genes in O-GlcNAc proteome, c = genes associated with the phenotype term/human ID gene orthologs, d = background/all *Drosophila* genes. Significance was determined using two-sided Fisher's exact test in R [95]. p-values were adjusted for multiple testing ($p_{adj}$) with Bonferroni-Holm correction [96].

### Supporting information

**S1 Fig. Generation and characterization of $sxc^{H596F}$, $sxc^{L283F}$, $sxc^{R313P}$ and $sxc^{A348T}$ alleles.** (A) Schematic representation of *Drosophila* sxc protein showing the location of H596F, L283F, R313P and A348T mutations; purple tetratricopeptide repeat (TPR) domain, green glycosyl transferase (GT) domain. **(B)–(E)** Sequences of genomic DNA of wild type, sxcH596F, sxcL283F, sxcR313P and sxcA348T *Drosophila* alleles. The missense mutation and additional

silent mutations are highlighted. The restriction digestion sites used for genotyping are shown.
(TIF)

**S2 Fig. Jump responses in the fatigue assay.** In the fatigue assay, jump responses were induced with 50 light-off pulses with 5 s interval between pulses that prevents habituation. The jump response is presented as % of jumping flies in each light-off trial. The mean number of trials that flies needed to reach the no-jump criterion (Trials To Criterion, TTC) ± SEM is presented. (A) Jump response of the $sxc^{H537A/+}$; $Oga^{KO/+}$ flies (N = 85, mean TTC ± SD: 27.4 ± 8, in cyan) remains high throughout the entire course of the experiment, similar to control flies (+/+, N = 85, mean TTC ± SD: 34.5 ± 5, $p_{adj}$ = 0.14, in blue) demonstrating that restored habituation in $sxc^{H537A/+}$; $Oga^{KO/+}$ flies (**Fig 1B**) is not confounded by fatigue. (B) Jump response of the $sxc^{H537A/+}$; *elav-Gal4>UAS-sxc* flies (N = 52, mean TTC ± SD: 25.1 ± 7.9, in green) remains high throughout the entire course of the experiment, similar to control flies (+/+, N = 55, mean TTC ± SD: 28.3 ± 2.3, $p_{adj}$ = 0.128, in blue) demonstrating that restored habituation in $sxc^{H537A/+}$; *elav-Gal4>UAS-sxc* flies (**Fig 1D**) is not confounded by fatigue. (C) Jump response of the $sxc^{R313P/+}$; $Oga^{KO/+}$ flies (N = 73, mean TTC ± SD: 28.9 ± 7.6, in cyan) remains high throughout the entire course of the experiment, similar to control flies (+/+, N = 84, mean TTC ± SD: 26.1 ± 5.1, $p_{adj}$ = 1, in blue) demonstrating that restored habituation in $sxc^{R313P/+}$; $Oga^{KO/+}$ flies (**Fig 4C**) is not confounded by fatigue. (D) Jump response of the $sxc^{A348T/A348T}$; $Oga^{KO/KO}$ flies (N = 78, mean TTC ± SD: 26.9 ± 5.1, in cyan) remains high throughout the entire course of the experiment, similar to control flies (+/+, N = 85, mean TTC ± SD: 34.5 ± 5, $p_{adj}$ = 0.31, in blue) demonstrating that restored habituation in $sxc^{A348T/A348T}$; $Oga^{KO/KO}$ flies (**Fig 4D**) is not confounded by fatigue. (E) Jump response of the $sxc^{R313P/+}$; $Oga^{D133N/+}$ flies (N = 83, mean TTC ± SD: 25.5 ± 9.2, in cyan) remains high throughout the entire course of the experiment, similar to control flies (+/+, N = 84, mean TTC ± SD: 26.1 ± 5.1, $p_{adj}$ = 1, in blue) demonstrating that restored habituation in $sxc^{R313P/+}$; $Oga^{D133N/+}$ flies (**Fig 4E**) is not confounded by fatigue. (F) Jump response of the $sxc^{A348T/A348T}$; $Oga^{D133N/D133N}$ flies (N = 63, mean TTC ± SD: 24.4 ± 5.1, in cyan) remains high throughout the entire course of the experiment, similar to control flies (+/+, N = 85, mean TTC ± SD: 34.5 ± 5, $p_{adj}$ = 0.15, in blue) demonstrating that restored habituation in $sxc^{A348T/A348T}$; $Oga^{D133N/D133N}$ flies (**Fig 4F**) is not confounded by fatigue. * $p_{adj}$<0.1, based on lm analysis with Bonferroni-Holm correction for multiple comparisons. Complete list of p-values and summary statistics is provided in **S3 Table**.
(TIF)

**S3 Fig. NMJ of $sxc^{H537A/H537A}$ mutant and NMJ branching morphology.** (A) Number of synaptic branches and branching points in $sxc^{H537A/+}$ larvae (N = 29, in red) is not significantly different from the genetic background control larvae (+/+, N = 24, branches: $p$ = 0.1439, branching points: $p$ = 0.05648, in blue). P-values are based non-parametric Wilcoxon test analysis. (B) Number of branches and branching points is not affected in *elav-Gal4>UAS-sxc* larvae (N = 29, in dark blue) compared the *elav-Gal4/+* larvae (N = 30, branches: $p_{adj}$ = 0.8702, branching points: $p_{adj}$ = 0.8488, in light blue) and to *UAS-sxc/+* larvae (N = 26, branches: $p_{adj}$ = 0.8294, branching points: $p_{adj}$ = 0.2689, in grey). (C) Branches and branching points are not affected in $sxc^{H537A/+}$; *UAS-sxc/+* larvae (N = 28, in red) compared to the control larvae (+/+, N = 28, branches: $p_{adj}$ = 0.7121, branching points: $p_{adj}$ = 0.2979, in blue). $sxc^{H537A/+}$; *elav-Gal4>UAS-sxc* larvae (N = 29, in green) do not show any changes in number of branches and branching points compared to the $sxc^{H537A/+}$; *UAS-sxc/+* larvae (branches: $p_{adj}$ = 0.4097, branching points: $p_{adj}$ = 0.5928) and control larvae (+/+; branches: $p_{adj}$ = 0.4301, branching points: $p_{adj}$ = 0.6927). (D) $sxc^{H537A/H537A}$ larvae have significantly increased NMJ length (N = 25, $p$ =) and perimeter (N = 21, $p$ =, in red) compared to their genetic background control (+/+, N = 26, in blue) but not significantly different number of boutons ($p$ = 0.085), NMJ area

($p = 0.618$), number of branches ($p = 0.691$) and branching points ($p = 0.371$). * $p<0.05$, ***
$p<0.001$. P-values for boutons, length, area and perimeter are based on one-way ANOVA. P-
values for branches and branching points are based on non-parametric Wilcoxon test analysis.
(E) Branches and branching points are not affected in $sxc^{H956F}$ larvae (N = 31, branches: $p_{adj}$ =
1, branching points: $p_{adj}$ = 0.5, in brown), $Oga^{KO}$ larvae (N = 30, branches: $p_{adj}$ = 0.75, branch-
ing points: $p_{adj}$ = 0.51, in purple), and $sxc^{H596F}$; $Oga^{KO}$ larvae (N = 30, branches: $p_{adj}$ = 0.75,
branching points: $p_{adj}$ = 0.5, in cyan) compared to the genetic background control larvae (+/+,
N = 28, in blue). $sxc^{H596F}$; $Oga^{KO}$ larvae do not show a significant change in number of
branches and branching points compared to the $sxc^{H956F}$ larvae (branches: $p_{adj}$ = 0.75, branch-
ing points: $p_{adj}$ = 1). Data presented as individual data points with mean ± SD. P-values are
based on Kruskal-Wallis test with Wilcoxon pairwise test for multiple comparisons. Complete
list of p-values and summary statistics is provided in S3 Table. (D') Representative NMJs of
genetic background control (+/+) and $sxc^{H537A/sxcH537A}$ wandering third instar larvae labeled
with anti-discs large 1 (*Dlg*, magenta) and anti-synaptotagmin (*Syt*, green). *Scale bar*, 20μm.
The quantitative parameter values of the representative images (+/+ | $sxc^{H537A/sxcH537A}$): #Bou-
tons (26 | 30), Length (96.2 | 169.8), Area (415.9 | 464.3), Perimeter (258.4 | 445).
(TIF)

**S4 Fig. Western Blot and developmental survival of $sxc^{H596F}$ flies.** (A) Embryos from either
wildtype, $sxc^{H537A}$, $sxc^{H596F}$ homozygotes were assessed for levels of global O-GlcNAc using a
pan-O-GlcNAc antibody RL2. The blot was normalized to actin. This blot is a representative
of three experiments. (B) Reduced total O-GlcNAc levels in $sxc^{H596F}$ and $sxc^{H537A}$ homozygotes
are not associated with developmental lethality. Data presented as percentage of pupae and
adults derived from stage 11–16 embryos (100 per genotype per experiment, n = 3). Based on
Student's t-test with Holm-Sidak's correction for multiple testing. Complete list of p-values
and summary statistics is provided in S3 Table.
(TIF)

**S5 Fig. Developmental and locomotor characterization and NMJ branching morphology
of $sxc^{R313P}$, $sxc^{A348T}$ and $sxc^{L283F}$.** (A) Control, $sxc^{R313P}$, $sxc^{A348T}$ and $sxc^{L283F}$ embryos (stage
11–16, 80–100 per experiment) were transferred to fresh food at 25˚C, and the numbers of
pupae formed and adults eclosed were counted. Development from embryo to pupae or from
pupae to adulthood was not significantly affected in $sxc^{R313P}$ (pupae: N = 4 repeats, p = 0.9,
adults: N = 3, p = 0.29, in red), $sxc^{A348T}$ (pupae: N = 3, p = 0.152, adults: N = 3, p = 0.345, in
blue) and $sxc^{L283F}$ mutants (pupae: N = 4, p = 0.108, adults: N = 3, p = 0.727, in green). Data
presented as individual data points with mean ± SD. P-values are based on Student's t-test. (B)
Flight escape performance was assessed in the island assay. 15 flies per measurement were
thrown on a white platform surrounded with water. Data was collected over 3 days of measure-
ment (Control: N = 23, $sxc^{R313P}$: N = 16, $sxc^{A348T}$: N = 15 and $sxc^{L283F}$: N = 14 repeats). Floating
bars depict mean ± SD area under curve (AUC), a parameter that is derived from data plotted
as % flies on the platform over time. One-way ANOVA with Tukey's multiple comparisons
was used to compare the mean AUC between genotypes. Flight escape performance of
$sxc^{R313P}$, $sxc^{A348T}$ and $sxc^{L283F}$ flies revealed no defects in locomotion or fitness. (C) Number of
synaptic branches is increased in $sxc^{R313P}$ (N = 21, p = 0.049, in red) and $sxc^{L283F}$ larvae
(N = 20, in green) compared to the genetic background control (+/+, N = 41, in grey). Number
of branching points is not significantly different. Data presented as individual data points with
mean ± SD. * p< 0.05. P-values are based on Kruskal-Wallis test with Wilcoxon pairwise test
for multiple comparisons. Complete list of p-values and summary statistics is provided in S3
Table.
(TIF)

**S1 Table. Primer sequences.**
(XLSX)

**S2 Table. Light-off jump habituation parameters summary.**
(XLSX)

**S3 Table. Summary statistics.**
(XLSX)

**S4 Table. Enrichment in *Drosophila* phenotypes and orthologs of human genes implicated in intellectual disability.**
(XLSX)

# Acknowledgments

We thank Jennifer Milligan for help with the Western blots and Mehmet Gundogdu for help with the transgenic strains.

# Author Contributions

**Conceptualization:** Michaela Fenckova, Villo Muha, Daniel Mariappa, Andrew T. Ferenbach, Annette Schenck, Daan M. F. van Aalten.

**Data curation:** Michaela Fenckova, Villo Muha, Daniel Mariappa, Marica Catinozzi, Ignacy Czajewski, Laura E. R. Blok, Andrew T. Ferenbach.

**Formal analysis:** Michaela Fenckova, Villo Muha, Daniel Mariappa, Marica Catinozzi, Ignacy Czajewski, Laura E. R. Blok, Andrew T. Ferenbach.

**Funding acquisition:** Erik Storkebaum, Annette Schenck, Daan M. F. van Aalten.

**Investigation:** Michaela Fenckova, Villo Muha, Daniel Mariappa, Marica Catinozzi, Ignacy Czajewski, Laura E. R. Blok.

**Methodology:** Michaela Fenckova, Villo Muha, Daniel Mariappa, Marica Catinozzi, Andrew T. Ferenbach.

**Project administration:** Michaela Fenckova, Annette Schenck, Daan M. F. van Aalten.

**Resources:** Andrew T. Ferenbach, Annette Schenck, Daan M. F. van Aalten.

**Supervision:** Erik Storkebaum, Annette Schenck, Daan M. F. van Aalten.

**Visualization:** Michaela Fenckova, Villo Muha, Daniel Mariappa.

**Writing – original draft:** Michaela Fenckova, Villo Muha, Daniel Mariappa, Annette Schenck, Daan M. F. van Aalten.

**Writing – review & editing:** Michaela Fenckova, Villo Muha, Daniel Mariappa, Marica Catinozzi, Ignacy Czajewski, Erik Storkebaum, Annette Schenck, Daan M. F. van Aalten.

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
