## [Decision Letter · Decision Letter 0]

2 Sep 2021

Dear Dr van Aalten,

Thank you very much for submitting your Research Article entitled 'Intellectual disability-associated disruption of O-GlcNAcylation impairs neuronal development and cognitive function in Drosophila' to PLOS Genetics.

The manuscript was fully evaluated at the editorial level and by four independent peer reviewers who are experts in the field. As you can see from the reports below, the reviewers acknowledged the importance of this work and appreciated the authors' efforts in generating mutant fly lines for OGT and designing experiments for testing neuronal and cognitive effects. However, the reviewers also raised substantial concerns, including comments related to analysis of null allele, testing patient-specific mutants for synaptic assays, clarity and criteria for terminology used in the paper, and conclusions drawn from bioinformatic analysis towards neural processes and function. Based on the reviews, we will not be able to accept this version of the manuscript, but we would be willing to review a much-revised version. We cannot, of course, promise publication at that time.

If you decide to revise the manuscript for further consideration at PLOS Genetics, please aim to resubmit within the next 60 days, unless it will take extra time to address the concerns of the reviewers, in which case we would appreciate an expected resubmission date by email to plosgenetics@plos.org.

[LINK]

We are sorry that we cannot be more positive about your manuscript at this stage. Please do not hesitate to contact us if you have any concerns or questions.

Yours sincerely,

Santhosh Girirajan

Associate Editor

PLOS Genetics

Gregory P. Copenhaver

Editor-in-Chief

PLOS Genetics

Reviewer's Responses to Questions

**Comments to the Authors:**

**Reviewer #1**: The manuscript by Fenckova et al explores cognitive mechanisms in the fly focusing on mutations identified in humans as causative of intellectual disability in the gene O-GlcNAc transferase (OGT). The habituation studies are carefully crafted with well controlled genetic tools. The authors offer important genetic evidence supporting a model where by disruption of the O-GlcNAc addition-removal cycle is causative of cognitive dysfunction in Drosophila models of human mutations. The most clear cut and conclusive evidence is emerging of the Drosophila habituation assays presented in Figures 1 and 4. In my view, these genetic-behavioral experiments provide a solid foundation for the assertion in the title that “Intellectual disability-associated disruption of O-GlcNAcylation impairs […] cognitive function in Drosophila”. However, the paper falls short to provide a comprehensive understanding of processes affected by the diverse mutations tested either individually or collectively. It is clear that global defects on O-GlcNAc-sylation do not account for the habituation phenotypes described in figure 4. Yet, the identity of neural processes and targets are insufficiently explored with a limited data-mining analysis of the Drosophila O-GlcNAc proteome. This last part of the paper is truncated and inconclusive.

Concerning figures 2 and 3, in my view they need some reconsideration. Figure 3 is a collection of important controls that can go into supplementary materials.

Figure 2, while I can see the graphs depicting the reported differences in boutons number and other NMJ parameters, I cannot see in the images the phenotypes reported by the graphs. Thus, I cannot concur with the authors that ”Intellectual disability-associated disruption of O-GlcNAcylation impairs neuronal development […]”. This assertion needs better NMJ evidence and/or new approach to document this conclusion.

Appendix has references at the end not related to the content of the paper.

In summary, only two figures provide uncontestable insight (Fig. 1 and 4) making this manuscript a solid yet rather discrete contribution.

**Reviewer #2**: Review of PGenetics-D-21-00961

Summary

Overall I find this is an interesting study using Drosophila as a model system to understand the function of O-GlyNAcylation on neuronal development and function. I further like the use of CRISPR to generate specific mutant alleles found in human clinical subjects and thus explore the mechanism of specific allelic variants in human disease.

I also think the analysis well planned and executed. My comments and criticisms are minor but I offer those for consideration of the author and editor.

Questions and concerns (not in any particular order)

1. I do think that labeling habituation responses as cognitive function will not be well received by most human geneticists. There has always been a tension between model organism researchers and human geneticists regarding when invertebrate models reflect human pathological mechanisms. I would advise not using this term, but simply stating that they have found affects on habituation and synapse development.

2. Have they done any CNS morphological assessments of their mutants?

3. I am a bit puzzled by their results with the sxc[H537A] allele. As heterozygotes this allele affects bouton number but not other parameters of synapse size. Yet, when the homozygote is analyzed it displays phenotypes for NMJ length and perimeters but not bouton number. Likewise I noticed that the effect of this allele as a heterozygote on bouton number is a bit modest, and show some variation within experiments shown (for example see the difference between Fig. 2 Panel A, and Figure 2 Panel C. The difference between sxc/+ and +/+ is pretty modest in Panel C, affected largely by a few outliers. I think some further discussion of these findings is warranted. In my mind it tells me the effect of synapse morphology is modest and makes me wonder if there are any observable electrophysiological changes.

4. I found the allele specific analysis very interesting and followed their stepwise arrival at the conclusion that while overall GlycNAcylation was not changed in some alleles, specific protein targets do likely see differences in modification since the phenotypes can be rescued by altering Oga function. However, I am not a big fan of making biological conclusions from multi-step bioinformatics analyses, and while their enrichment analysis seems valid, it is a far cry from doing an experiment. Their ability to look at the patterns of GlycNAcylation by western seems a logical place to begin to see if different alleles give different patterns (ie intensities) of modification. I would be intrigued if this sort of analysis was attempted and if there were any indications of differential modification of protein targets.

**Reviewer #3**: In this manuscript, Fenckova et al., studies the role of OGT (O-GlcNAc transferase) and disease associated variants in this gene using Drosophila melanogaster. First the authors studied a previously reported hypomorphic allele of sxc (OGT fly ortholog with H537A mutation) that has diminished enzymatic activity and showed that heterozygous mutants show defects in a light-induced habituation assay (and the homozygous are defective in jumping in response to light stimuli). This phenotype can be rescued by re-introducing sxc in neurons, or via removing one copy of Oga, a gene that encodes the O-GlcNAc hydrolase. This indicates that this gene is required in the neurons for proper habituation and also suggests that proper O-GlcNAc cycling (or dynamics) is required for this behavior. The authors also showed that hypomorphic sxc mutants (tested H537A and H596F) show altered synaptic morphology at the larval neuromuscular junction and that this phenotype can also be rescued by the same genetic manipulations. Interestingly, over-expression of sxc also showed synaptic defects, suggesting that tight control of protein O-GlcNAc modification is critical for proper synaptic morphology. Next, the authors generated knock-in flies that carry mutations that corresponds to one of three previously reported OGT alleles found in patients (R313P, A348T, L283F) and subjected them to biochemical, behavioral and histological assays. Through western blotting of O-GlcNACylated proteins, the authors found that these mutants do not seem to have a general/global defect in protein O-GlcNACylation. The authors showed that A348T and L283F mutants did not show a major motor defect but the R313P did show some motor coordination defect through a negative geotaxis assay. Because mutant alleles of sxc that show strong enzymatic defect do not show this defect, the authors interpreted this as a sign that the R313P variant causes some sort of an enzymatic activity independent function. Next, they tested the light-induced habituation in the three patient alleles and found that two of them show a significant phenotype which can be corrected by reducing Oga (L283F line were not good responders to light stimuli, despite their normal motor behaviors in other tests). Finally, the authors performed a bioinformatics analysis on the Drosophila O-GlcNAc proteome and found that proteins encoded by genes involved in nervous system function and homologs of human ID genes are enriched in this group. The authors list 9 high-confidence mapped O-GclNAc genes that are homologous to human ID genes and propose that these maybe potential targets that may be responsible for the phenotypes described in this study.

I feel the authors have put in significant effort to generate a number of fly knock-in strains to functionally characterize the role sxc/OGT and disease associated variants in a cognition related behavior and synaptic morphology. I feel this work will be of interest to the readership of PLoS Genetics, especially to fly researchers interested in neuronal function, glycobiologists who study physiological significance of certain sugar modifications, and clinicians and human geneticists who study disease associated variants. I would be happy to recommend this study for publication, if the following points are addressed.

Major Points

1) The author performs the habituation assay on a number of sxc mutants but do not show data for the null allele. I acknowledge the homozygous null flies are pupal lethal but the authors should be able to test this in a heterozygous fly. Since the null allele is an important reference point to interpret the function of missense alleles (e.g. if the mutant phenotype of certain missense allele is stronger than the null, they maybe antimorphic or neomorphic alleles, rather than a hypomorph), it would be valuable to show what the null allele looks like in this assay (which have been generated by the authors in PMID: 29588363. In addition, multiple alleles are available from stock centers). Also, note that the habituation data for the strong hypomorphic allele (H596F) they generated is also not provided here, which would make this paper more complete.

2) The authors explored the synaptic morphology phenotype of the known enzymatic defective mutants (H537A and H596F), but have not examined this for the patient derived variants (R313P, A348T, L283F). Was this because they looked at these phenotypes and did not see any relevant phenotype? Or was there a reason this experiment cannot be carried out? Since the authors are trying to make the point that these synaptic phenotype has some relevance to human ID, it would make sense to show how the NMJs of the flies with mutations that are analogous to those found in the patient mutations look like. Also, data from the heterozygous or homozygous null allele can also be provided here to strengthen this paper.

3) The authors attribute the negative geotaxis defect of the R313P allele to this variant affecting a non-enzymatic function of Sxc. While this is an interesting idea, alternatively this phenotype may be due to some sort of a 2nd site mutation that is unique to this strain (off targeting by CRISPR or some floating mutation that is closely linked that was not eliminated through back crossing). The authors should perform a rescue experiment (e.g. neuronal rescue experiment using UAS/GAL4 as they do for other alleles or genomic rescue) if they wish to claim that the negative geotaxsis defect in this allele is due to sxc and not other genes.

4) The section subtitle “Patient-related sxc mutant alleles have normal levels of O-GlcNAcylation” seems a bit misleading, especially since they later argue that phenotypes seen in some of the patient mutant alleles can be corrected by removing one copy of Oga. As they discuss in the text that there may be some minor O-GlcNAcylatiojn defect that is not obvious when performing a bulk O-GlcNAcylation assay by western blot (e.g. a few substrates that are more sensitive to slight reduction in O-GlcNAcylation is responsible for the habituation and synaptic phenotypes). The authors may want to consider an alternative title like “Patient-related sxc mutant alleles do not affect global protein O-GlcNAcylation”.

Minor Points:

1) This last sentence of the abstract (“This study establishes a critical role for O-GlcNAc cycling and disrupted O-GlcNAc transferase activity in cognitive dysfunction and intellectual disability and points to potential treatment strategy for OGT-CDG.”) is a bit length with many “and” being used. Better to split into two sentences or rephrase.

2) The fly ortholog of OGT (sxc) has been reported to be a member of the Polycomb group as the authors discuss. Considering that many members of Polycomb genes are associated with human diseases that affect the nervous system including ID (reviewed in papers such as PMID: 34426021), the author may want to discuss the potential connection between OGT, Polycomb genes and O-GlcNAcylation, if there may be a link.

3) There is one prior study that found a role for OGT in the nervous system, specifically in the context of circadian rhythm regulation using timeless-GAL4 and UAS-RNAi. Although this study may not be directly relevant to this work, the author may want to consider citing this it as one previous evidence in flies that this gene has critical functions in the nervous system (note that this reviewer is not the author of this study nor have any conflict of interest).

**Reviewer #4**: Manuscript #: PGENETICS-D21-00961

Title: Intellectual disability-associated disruption of O-GlcNAcylation impairs neuronal development and cognitive function in Drosophila.

Summary: Fenckova et al. present an impressive series of behavioral, genetic, and molecular experiments to uncover phenotypes caused by increased and decreased O-GlcNAc modification. In addition to the use of existing genetic tools, they develop and molecularly characterize new strains that both further decrease transferase activity and mimic the intellectual disability (ID)-associated missense mutations found in humans. The focus of their experimental work centers on a simple learning assay, an habituated jump in response to repeated lights-off stimuli. The authors then extend these findings by hypothesizing a mechanistic basis, synaptic change, and by using additional behavioral assays. The experimental findings are, for the most part, clear and the text is well-written. While I recommend it for publication, I strongly suggest a number of major and minor revisions that should provide for more clarity and a greater over-all impact.

Major Revisions:

1. In both the title and throughout the Introduction and Discussion, the authors use the term ‘cognitive function’ lightly, basing this on the habituation defects observed in their mutants. However, cognition generally describes more complex, executive functioning and its use may misrepresent the experimental findings. A more fitting word choice would simply be ‘habituation’.

2. In the learning and memory literature, habituation as a form of learning is accepted once a number of criteria are met. First, the decreased response should not be fatigue (which the authors effectively rule out) and sensory adaptation should not explain the results (this is unaddressed by the authors). To suggest learning, the habituated response should also show a) recovery by dishabituating stimuli (an air puff, for example), b) that the jump response recovers spontaneously in a stimulus-dependent fashion, and c) that the habituation is stimulus specific. The authors should address (experimentally or through literature review) at least one of these criteria (a-c) before claiming that learning is being measured. If they or others have shown that sensory adaptation does not explain the habitutation, that too should be documented.

3. While the Introduction was informative as to the importance of O-GlcNAcylation, the authors make frequent interpretations of their habituation assay and of sxc function in the Results that are only explained after reading the Discussion, and then only briefly. In particular, the reader would benefit from a short introduction to a) habituation learning, b) the function of the TPR domain of sxc in relation to the glycosyl transferase catalytic region, and c) a structure-function analysis of ID-specific mutations (through crystallography), and their known effects (if any) on substrate interactions.

4. Regarding the light-off jump habituation, the data indicates that not only do genetic manipulations of O-GlcNAcylation alter the rate of habituation, but they also influence the post-habituation baseline activity level (indicated by higher baselines in Figure 1 and Figure 4). Furthermore, the rescue of the trials to criterion (TTC) throughout the paper correlates with the return of this baseline activity back to control levels. The authors should explain their interpretation of this higher baseline at its first occurrence in the Results. I am also wondering if, since the authors are measuring wing vibration and the actual jump frequency, there may be a change in the animal’s excitability. Given the change in NMJ bouton number (increased in some genotypes), a change in neuronal excitability might also be hypothesized in the Discussion.'

Minor Revisions:

1. Throughout the text there numerous grammatical errors (missing ‘the’, ‘for’, ‘a’, for example).

2. On page 6, lines 123 and 124, and then again on page 7, lines 148 and 149, the authors present data without showing the n or any statistics. These additions will strengthen the author’s conclusions and should be included.

3. The first sentence of the paragraph at the bottom of page 12 (lines 288-291) is a run-on and should be broken into two.

4. At the end of page 15 and the top of page 16 of the Discussion the authors discuss the link between O-GlycNAcylation and microtubule polymerization, but they don’t indicate the details that may be relevant – mechanisms of the modulation, for example, which may be relevant to the NMJ length measurements.

5. In Figure 2 A’-D’, larval NMJ synapses are labeled with a pre- and post-synaptic marker. There are also arrowheads to distinguish the type 1b synapses from other synapses. However, there appear only type 1b boutons in the images, making the arrowheads unnecessary.

**Have all data underlying the figures and results presented in the manuscript been provided?**

Reviewer #1: Yes

Reviewer #2: Yes

Reviewer #3: Yes

Reviewer #4: Yes

PLOS authors have the option to publish the peer review history of their article (what does this mean?). If published, this will include your full peer review and any attached files.

Reviewer #1: No

Reviewer #2: No

Reviewer #3: No

Reviewer #4: No

---

## [Decision Letter · Decision Letter 1]

21 Mar 2022

Dear Dr van Aalten,

We are pleased to inform you that your manuscript entitled "Intellectual disability-associated disruption of O-GlcNAcylation impairs neuronal development and cognition-relevant habituation learning in Drosophila" has been editorially accepted for publication in PLOS Genetics. Congratulations!

Please note that Reviewer 1 has suggested to use the short title as the main title of the paper.  If the authors are in favor of this suggestion, we ask that this change is made in the final version of the manuscript.

Yours sincerely,

Santhosh Girirajan

Associate Editor

PLOS Genetics

Gregory P. Copenhaver

Editor-in-Chief

PLOS Genetics

Comments from the reviewers (if applicable):

Reviewer's Responses to Questions

**Comments to the Authors:**

Reviewer #1: The authors have done a thorough and comprehensive job to amend the paper. Now it reads nicely and presents a solid case for the conclusions reached by the authors.

The short title is much better than the long title, I suggest the authors use the short title as the main title of this paper.

Reviewer #3: The authors have made significant efforts to address my earlier concerns and questions. I am happy to support the publication of this paper in PLoS Genetics.

Reviewer #4: No response

**Have all data underlying the figures and results presented in the manuscript been provided?**

Reviewer #1: Yes

Reviewer #3: Yes

Reviewer #4: Yes

PLOS authors have the option to publish the peer review history of their article (what does this mean?). If published, this will include your full peer review and any attached files.

Reviewer #1: **Yes: **Victor Faundez

Reviewer #3: No

Reviewer #4: No

**Data Deposition**

http://datadryad.org/submit?journalID=pgenetics&manu=PGENETICS-D-21-00961R1

**Press Queries**

---

## [Editor Report · Acceptance letter]

25 Apr 2022

PGENETICS-D-21-00961R1 

Intellectual disability-associated disruption of O-GlcNAc cycling impairs habituation learning in Drosophila  

Dear Dr van Aalten, 

We are pleased to inform you that your manuscript entitled "Intellectual disability-associated disruption of O-GlcNAc cycling impairs habituation learning in Drosophila " has been formally accepted for publication in PLOS Genetics! Your manuscript is now with our production department and you will be notified of the publication date in due course.

With kind regards,

Agnes Pap

PLOS Genetics

On behalf of:
